# Extensive standing genetic variation from a small number of founders enables rapid adaptation in *Daphnia*

Anurag Chaturvedi [1,2,3,10 ✉], Jiarui Zhou [3,4,10], Joost A. M. Raeymaekers [5,6], Till Czypionka[1], Luisa Orsini [3], Craig E. Jackson[7], Katina I. Spanier [1], Joseph R. Shaw[7], John K. Colbourne[3] & Luc De Meester[1,8,9]

We lack a thorough understanding of the origin and maintenance of standing genetic variation that enables rapid evolutionary responses of natural populations. Whole genome sequencing of a resurrected *Daphnia* population shows that standing genetic variation in over 500 genes follows an evolutionary trajectory that parallels the pronounced and rapid adaptive evolution of multiple traits in response to predator-driven natural selection and its subsequent relaxation. Genetic variation carried by only five founding individuals from the regional genotype pool is shown to suffice at enabling the observed evolution. Our results provide insight on how natural populations can acquire the genomic variation, through colonization by a few regional genotypes, that fuels rapid evolution in response to strong selection pressures. While these evolutionary responses in our study population involved hundreds of genes, we observed no evidence of genetic erosion.

[1] Laboratory of Aquatic Ecology, Evolution and Conservation, KU Leuven, Leuven, Belgium. [2] Department of Ecology and Evolution, University of Lausanne, Lausanne, Switzerland. [3] Environmental Genomics Group, School of Biosciences, University of Birmingham, Birmingham, UK. [4] Centre for Computational Biology, University of Birmingham, Birmingham, UK. [5] Laboratory of Biodiversity and Evolutionary Genomics, KU Leuven, Leuven, Belgium. [6] Faculty of Bioscience and Aquaculture, Nord University, Bodø, Norway. [7] O'Neill School of Public and Environmental Affairs, Indiana University, Bloomington, IN, USA. [8] Leibniz Institut für Gewässerökologie und Binnenfischerei (IGB), Berlin, Germany. [9] Institute of Biology, Freie Universität Berlin, Berlin, Germany. [10]These authors equally to this work: Anurag Chaturvedi, Jiarui Zhou. ✉email: anurag.chaturvedi@unil.ch

Natural populations can adapt to environmental change within a few generations when there is sufficient genetic variation to support trait evolution in response to selection[1]. Such rapid trait change provides resilience to the elevated pace and magnitude of human-driven environmental change[2,3]. But there is much uncertainty on how this genetic variation is acquired, structured, and maintained, and how precisely natural selection induces genome-wide changes when multiple loci or genes are involved.

Complex adaptive traits are assumed to be shaped and to evolve through changes in many loci. We know from theory that if change in the frequency of a beneficial allele under positive selection is more rapid than the regional recombination rate, the variation within genomic regions around the selected locus will be distorted by hitchhiking on linked DNA[4–6]. Such selective sweeps are responsible for detecting "footprints" of natural selection in genomes. Yet, they also conceal the identity of the actual loci targeted by selection, which can be revealed by witnessing changes over time.

Here, we directly quantified changes in genome-wide sequence variation in response to strong predator-driven selection and subsequent relaxation of that selection in a natural population of the waterflea Daphnia over time. The crustacean Daphnia is a keystone species in the food chains of lakes and ponds and an important model organism in ecology and evolution[7]. Its life cycle includes a stage of diapause during early development, producing a dormant stage that can be retrieved from dated lake sediments and resurrected to capture population-level genetic diversity through time[8].

In this work, by resequencing the genomes of resurrected genotypes over time of a natural Daphnia population that shows an elaborate evolutionary response to changes in fish predation pressure[9], we demonstrate (i) that the rapid evolutionary response observed in this population is largely based on standing genetic variation and (ii) that this standing genetic variation is maintained through time, irrespective of the responses to selection. By also sequencing genotypes from other spatially distributed populations that differ in terms of fish predation pressure, we also demonstrate (iii) that standing local genetic variation in the temporal population reflects regional standing genetic variation and (iv) that a few founding individuals from the regional genotype pool suffice to establish the extensive standing genetic variation fueling rapid evolution in natural populations of Daphnia.

## Results
We sequenced 36 whole genomes from three temporal subpopulations of Daphnia magna that persisted through a well-documented and pronounced change in fish predation pressure (Fig. 1a, see "Material and methods", Supplementary Table 1). The three temporal subpopulations were isolated from a man-made pond (Oud-Heverlee, Belgium) established in 1970. The pond was devoid of planktivorous fish during the first few years; in this period the invertebrate community naturally colonized the pond (pre-fish temporal subpopulation, 1970–1972). High densities of planktivorous fish were stocked between 1976 and 1979 (high-fish temporal subpopulation, 1976–1979). Fish stocking then gradually declined, leading to much lower fish densities in 1988 (reduced-fish temporal subpopulation, 1988–1990)[10]. Hence, the Daphnia within this pond in Oud-Heverlee experienced a very strong selection pressure and subsequent relaxation of selection by fish predation over a time period of only 16 years (Fig. 1a)[10]. The sequence data are used to compare the changing patterns of DNA variation over time to the results from an earlier quantitative genetic study on the life history and behavioral traits

of the same Daphnia genotypes used in our current study[9]. This earlier study showed that 13 out of 14 traits evolved in response to changes in predator selection pressure[9]. Most of these traits evolved to more predator-safe trait values in the high fish predation phase and partially reversed to the original state in the transition from high to low predation (Fig. 1b)[9].

**Directional change and reversal in allele frequencies**. We identified 724,321 single-nucleotide polymorphisms (SNPs) across the sequenced genomes of 36 individuals. The level of genetic differentiation between temporal subpopulations measured across all SNPs was low ($F_{ST}$ pre-fish vs high-fish = 0.027; $F_{ST}$ high-fish vs reduced-fish = 0.010) (Supplementary Fig. 1). The allele-frequency changes over time were not significant for 94.45% of all SNPs ($P$ value > 0.01, Waples test (Waples 1989))[11]. This result suggests that there is little-to-no contribution of genetic drift in structuring the genome-wide polymorphisms within this population, which is corroborated by the very large estimates of effective population size (Ne ~ 1.66 million) (Supplementary Table 2).

During the pre-fish to high-fish transition, 4.23% of the overall SNPs showed a significant change in allele frequency, while during the later transition from high-fish to relaxed-fish, 1.55% of SNPs showed significant allele-frequency changes (Waples test, $P$ value < 0.01). This result indicates that only a small fraction of the genome responded to predation selection, and that transition from pre-fish to high-fish (six years) induced twice as many changes compared with the transition from high-fish to reduced-fish (10 years) (Supplementary Fig. 1).

Because of the temporal design, our study provides a rare glimpse on how genome-wide polymorphisms are dynamically arranged by selection and its subsequent reversal. Of the 30,669 SNPs with changing nucleotide frequencies during the pre-fish to high-fish transition, 77.44% showed a reversal toward their ancestral frequency during the subsequent high-fish to reduced-fish transition (Fig. 2). This reversal was significant (i.e., frequency change is significant during both periods, but in opposite direction) for 1,753 (5.71%) of the SNPs (Waples test, $P$ value < 0.01; Fig. 2c). The number of reversals was significantly larger than chance reversals calculated over 10,000 runs of permutations[12] (number of reversals in permutations: 994; $P$ value < 1e-4). In contrast, only 18 (0.05%) of the SNPs showed a significant frequency change in the same direction across both transition periods over which we quantified evolution (Waples test, $P$ value < 0.01) (Fig. 2d).

**Distinguishing direct targets of selection vs hitchhiking**. The rapidly shifting allele frequencies across thousands of SNPs that mirror the response trajectory of evolved phenotypic traits (Fig. 1b) and align with the increase and subsequent relaxation in fish predation pressure indicate adaptive genomic responses (see supplementary text). However, because of genetic hitchhiking (increase in frequency due to linkage with a sweeping allele), not all genes with polymorphisms that show reversals are predicted to be the direct targets of natural selection. A characteristic "footprint" of strong selection over a short number of generations is the detection of broad islands of divergence that are the products of selective sweeps (Supplementary Table 3)[4]. These arise when strong selection results in a rapid increase in the frequency of beneficial alleles and ancestral variation from nearby region hitchhikes along with these changes in the frequency of targeted alleles[4].

We determined that 80.20% of the SNPs that showed significant reversals were physically organized in genomic islands of high divergence during the first transition (582 islands

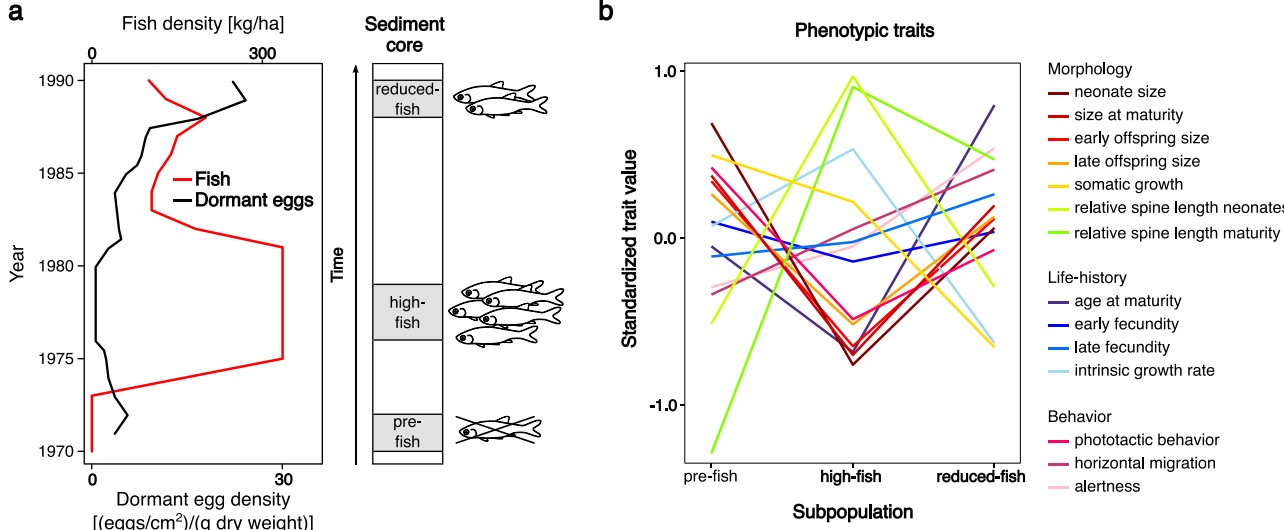

**Fig. 1 Resurrection ecology setting and trait change of *Daphnia magna* subpopulations separated by time in Oud-Heverlee Zuid (OHZ) pond, Belgium. a** Numbers of dormant eggs in relation to fish-stocking densities throughout the history of the pond (based on data from[10]). A schematic sediment core marks the time window of the "pre-fish period" (1970–1972), "high-fish period" (1976–1979), and "reduced-fish period" (1988–1990) temporal subpopulations. **b** Trait change across the three time periods for 14 morphological, behavioral and life-history traits, 10 of which showed a partial reversal in trait change when fish predation was relaxed and four of which changed in the same direction during the two transition periods (data from[9]). For clarity, error bars are not depicted (see[9] for more detail), but significant genetic changes in trait value were observed for all traits, except early fecundity. The observed genetic trait changes were in line with expectations under the assumption of adaptive evolution, suggesting that this population underwent a rapid (6- and 10-year time span) adaptive evolution in response to a strong increase and subsequent relaxation of fish predation pressure[9].

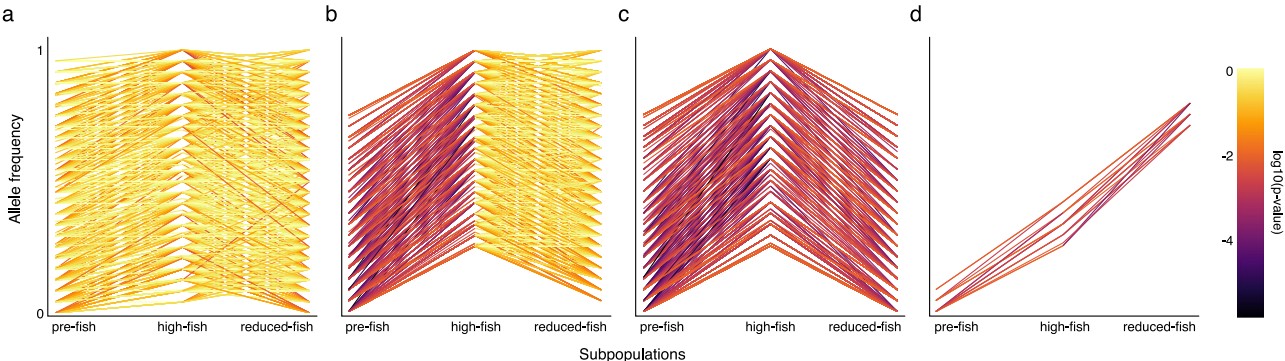

**Fig. 2 Allele-frequency changes at SNP variants through time in OHZ pond, Belgium. a** A random subset of 1000 SNPs showing no significant change in allele frequency over the entire period. **b** A random subset of 1000 outlier SNPs in the pre-fish to high-fish transition that exhibit no significant changes (Waples test, *P* values > 0.01) during the high-fish to the reduced-fish transition. Many SNPs show a tendency for a reversal in this data-set. **c** A random subset of 1000 out of the 1753 SNPs that show significant changes in allele frequency in the pre-fish to high-fish transition and also show a significant (Waples test, *P* value < 0.01) partial reversal of this change during the high-fish to reduced-fish transition. **d** All 18 SNPs are outlier SNPs in the pre-fish to high-fish transition and also show a further significant (Waples test, *P* value < 0.01) increase in allele frequency in the high-fish to reduced-fish transition. The color coding signifies the log10 significant changes in allele frequency for a specific transition (Waples test). Panels **a**, **b** and **c** are limited to 1000 SNPs to facilitate graphical display.

summing to 2.69 million nucleotides, i.e., approximately 2.69% of the genome) based on a hidden Markov model (HMM, Fig. 3a). These islands of high divergence are significantly larger in size (Mann–Whitney U test, upper-tailed, statistic = 173,517.5, *P* value = 9.78E-20) and on average contain a greater number of genes (Mann–Whitney U test, upper-tailed, statistic = 16,7619.5, *P* value = 2.16 E-4) than the islands of divergence identified for the second transition (Fig. 3b–c). While the reversal likely reflects the relaxation of selection, the reduction in size of the islands of divergence might reflect longer time for recombination to occur (i.e., 10 vs 6 years).

Tracking the dynamics of genomic islands and the action of recombination during the sequel of natural selection and its subsequent relaxation allows us to identify those potential regions of the genome that are differentiated because of positive selection vs linked responses during the first transition. We identified 390 genomic islands representing 1.93% of the genome in the pre-fish to high-fish transition that overlaps with 406 smaller genomic islands representing only 1.21% of the genome during the high-fish to reduced-fish transition (Fig. 3d). A total of 342 genes (2.79% of the total number of genes in *Daphnia*) were located in overlapping regions of these genomic islands (representing 0.83% of the *D. magna* genome), and are thus potentially under directional selection due to changes in predation pressure. This signal is quite reliable, given that the majority of overlapping highly differentiated genomic islands between the pre-fish to high-fish and the high-fish to reduced-fish transitions (i.e., 64.32%) only contain one gene. The remaining of overlapping

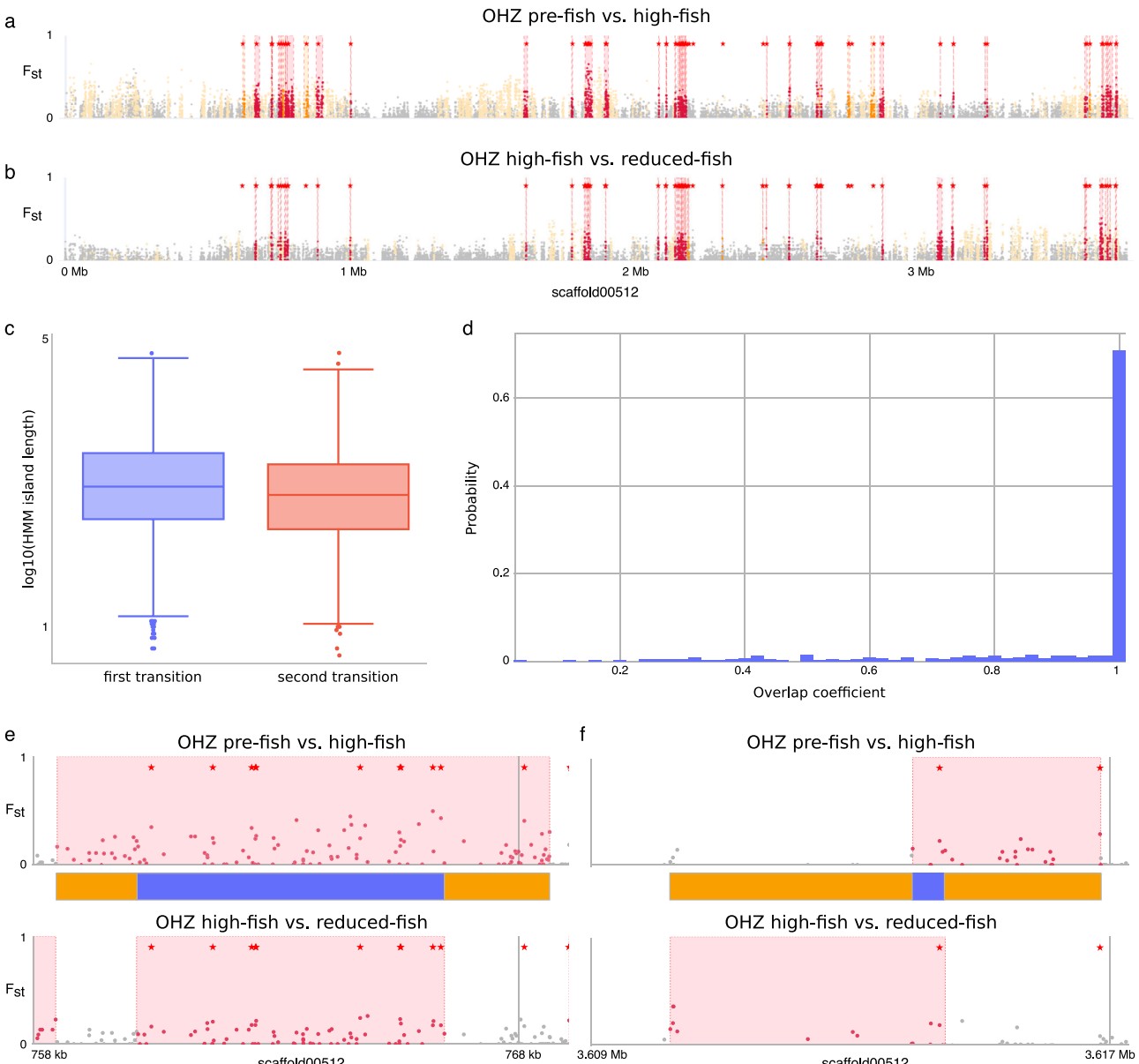

**Fig. 3 Hidden Markov model SNP categories of *Daphnia magna* temporal subpopulations separated by time in OHZ pond in Belgium. a** Genomic differentiation ($F_{ST}$) along scaffold 00512, the longest scaffold in the reassembled *D. magna* genome, between the pre-fish and the high-fish subpopulation. **b** Genomic differentiation ($F_{ST}$) along scaffold 00512 between the high-fish and the reduced-fish subpopulation. Patterns in other scaffolds are similar. **c** Box-plots showing the significant reduction (Mann–Whitney U test, upper tailed, statistic = 173,517.5, P value = 9.78E-20) in average length of HMM-detected islands of divergence between the pre-fish to high-fish ($n$ = 6111;minima: 0.70, maxima: 4.80, center: 2.95, Q1: 2.50, Q3: 3.41; lower fence: 1.15, upper fence: 4.73) and high-fish to reduced-fish ($n$ = 2879; minima: 0.60, maxima: 4.81, center: 2.83; Q1: 2.36, Q3: 3.26; lower fence: 1.04, upper fence: 4.58), outliers are defined as the data points more than three times of the interquartile range above Q3 or below Q1 transitions. **d** Histogram representing the overlap coefficient for all the overlapping HMM-detected islands of high divergence. The high values for the overlap coefficient suggest a high overlap between the islands of divergence between the two transitions. **e** An example illustrating the reduction in length of a HMM-detected island of divergence from the larger island of divergence in the pre-fish to high-fish transition (top) to a substantially smaller island in the high-fish to reduced-fish transition (bottom). **f** Example of reshuffling of a HMM-detected island of divergence in the pre-fish to high-fish transition (top) to the high-fish to reduced-fish transition (bottom). The stars in plots **a**, **b** and **e**, **f** indicate outlier SNPs. Red regions in plots **a**, **b** indicate overlapping islands of high divergence, whereas yellow regions indicate nonoverlapping (i.e., transition-specific) islands of high divergence. The pink boxes in plot **e**, **f** demarcate the HMM-detected island of divergence. The blue and orange boxes in plots **e**, **f** demarcate regions that harbor SNPs that are likely targeted by selection, or that are likely impacted by hitchhiking, respectively.

regions of highly differentiated genomic islands also harbor only two (16.7%) or three genes (9.1%). The nonoverlapping regions contain 131 genes (1.06% of genes in *Daphnia*), which likely hitchhiked with target genes of selection during the first transition. In other words, 72.30% of 473 detectable genes through the analysis of genomic islands of divergence harbor

reversible variation that is potentially targeted by natural selection (i.e., SNPs in the blue areas of Fig. 3e, f), while the allele-frequency trajectories of 27.70% of the genes associated with the islands of divergence during the first transition were likely impacted by hitchhiking (i.e., SNPs in the orange areas of Fig. 3e, f). The genes that were potentially direct targets of selection

(i.e., excluding hitchhiked genes) enriched for functionally recognizable biomolecular pathways (Supplementary Fig. 2), including the neuroactive ligand–receptor interaction pathway (Fisher exact test, adjusted $P$ value 7.13e-11), which is linked to phototactic behavior[9], the Wnt signaling pathway (Fisher exact test, adjusted $P$ value 2.64e-03), and the mTOR signaling pathway (Fisher exact test, adjusted $P$ value 1.09e-02). Phototactic behavior shapes diel vertical migration, a well-known fish-avoidance strategy that is known to have evolved in this population.

**Origin and maintenance of standing genetic variation.** All but six percent of the total genetic variation found in the genomes of the sequenced individuals across all three resurrected temporal subpopulations is detected in our sample of 12 individuals (24 genomes) of the founding (i.e., prefish) temporal subpopulation. This result suggests that the original founders were carrying all of the genetic variation that enabled the adaptive evolution of multiple traits in response to an increase and subsequent relaxation of fish predation pressure. The pre-fish temporal subpopulation is indeed highly polymorphic for these SNPs, with single individuals being polymorphic for 15–46% of all SNPs residing in the genomic islands that were potentially under directional selection when *Daphnia* faced increasing fish predation. Rarefaction analysis revealed that five randomly selected individuals from the prefish temporal subpopulation captured most of the polymorphisms, as they were on average polymorphic for 65% of the SNPs in the genomic islands (Fig. 4a). These percentages were even higher in the reduced-fish temporal subpopulation (24–53% of polymorphism for single individuals, on average 70% for five individuals, and up to 80% in the entire sample of 12 individuals; Fig. 4a). These results illustrate that the pronounced phase of selection, leading to evolutionary change for multiple traits[9] and genome-wide effects on allele frequencies, did not result in a reduction of genetic polymorphism, but rather in an increase in the amount of detected variation. None of the polymorphisms potentially under selection had reached genetic fixation within the fish-adapted population.

One plausible way in which standing genetic variation builds up in a local population is through immigration of genotypes from the surrounding landscape, either during colonization or through subsequent immigration. To assess regional genetic variation, we resequenced whole genomes of genotypes from twelve populations inhabiting ponds in the region (Flanders, Belgium), of which six harbor fish and six are devoid of fish (see Material and methods; Supplementary Figs. 1–4; Supplementary Table 1–3). A similar search for islands of divergence among these spatially distributed populations finds that 92% of the genes potentially under directional selection during the first transition (i.e., excluding hitchhiking regions) in our temporal population are similarly organized in islands of high genomic divergence in one or more of the spatial comparisons (Supplementary Fig. 4). Almost all of the biological pathways that are disproportionally represented by genes within islands of high divergence in the temporal set of populations overlap with pathways that are disproportionally represented in the spatial comparisons (Supplementary Fig. 2 and Supplementary Fig. 3). Given that local standing genetic variation matches regional variation, the question arises on how many individuals from the regional genotype pool were required to yield the observed initial genetic variation in our study population? A rarefaction analysis using randomly sampled individuals from populations in the region reveals that five random individuals from any of the other nine populations that were sampled in a periphery with radius of 10 km around the target pond harbored on average 80% of the

polymorphism in the potentially selected genomic regions of our target population (Fig. 4b, Supplementary Fig. 5). This result reveals that even a limited number of founding genotypes sampled from other populations in the surrounding landscape can result in a population with substantial genetic variation relevant with respect to the studied selection pressure.

## Discussion

Our study of a resurrected population, benchmarked against a spatial survey at the landscape level, reveals how natural populations can harbor substantial standing genetic variation that can be built up through colonization or immigration of just a few regional genotypes, and how standing genetic variation is structured through changing selection pressures over time. Polygenic predator-driven selection shaped approximately 1% of the *Daphnia* genome involving several hundreds of genes across multiple generations. The majority of potentially adaptive SNPs are distributed within genomic islands of high divergence that harbor almost 500 detectable genes, enriching a limited set of pathways. Natural selection likely accounts for the allele-frequency trajectories of the largest fraction of these genes, while the remaining genes (an estimated 28%) are hitchhiking with the targets of selection. The majority of alleles that showed significant frequency changes during a period of intense selection also showed a reversal after relaxation.

Patterns of adaptation to strong environmental gradients that are driven by a relatively small proportion of the genome but involving several hundreds of genes have been shown before in both vertebrates and invertebrates. For instance, freshwater adaptation in three-spined stickleback elicits responses involving <0.5–15% of the genome[13] and about 3% of the genes[14]. In another example, host races of the pea aphid living on alfalfa, clover, or pea differ in 9.12% of the genome and up to 8.10% of the genes[15]. While these examples represent major changes in habitat (marine to freshwater, host plant), the pattern of evolution documented in the present study represents a response to an in situ environmental change occurring during a time period of a few years, documented through data on whole genomes sampled through time. Size-selective predation by fish not only differs among habitats but can also rapidly change through time within the same habitat.

Our results show that the substantial genetic variation enabling the observed rapid genome-wide response to predation in our study population can be obtained from even a very limited number (e.g., as low as five) of founding individuals from the regional gene pool. Despite strong response to selection, we did not observe complete fixation of allelic variants, resulting in the maintenance of genetic variation. This lack of fixation might reflect that the periods of directional selection were relatively short in this population. Strikingly, however, we observed the same level of genetic polymorphism for SNPs under potential selection also in our spatial survey, both in populations with and without fish (Supplementary Fig. 6). This suggests that if the maintenance of genetic polymorphism is a reflection of short phases of directional selection, this is a common phenomenon across *Daphnia* populations. More likely the absence of genetic fixation might reflect the polygenic nature of the traits under selection, resulting in a relatively weak selection pressure on individual loci. Furthermore, our data suggest that the maintenance of high-standing genetic variation is also due to very large effective population sizes (>$1.10^6$), which are locally sustained by an extensive dormant egg bank, despite seasonal declines of *Daphnia* populations[16]. The combination of dormant egg banks and fluctuating environments further enhances the maintenance of standing genetic variation[17], which enables rapid evolution and

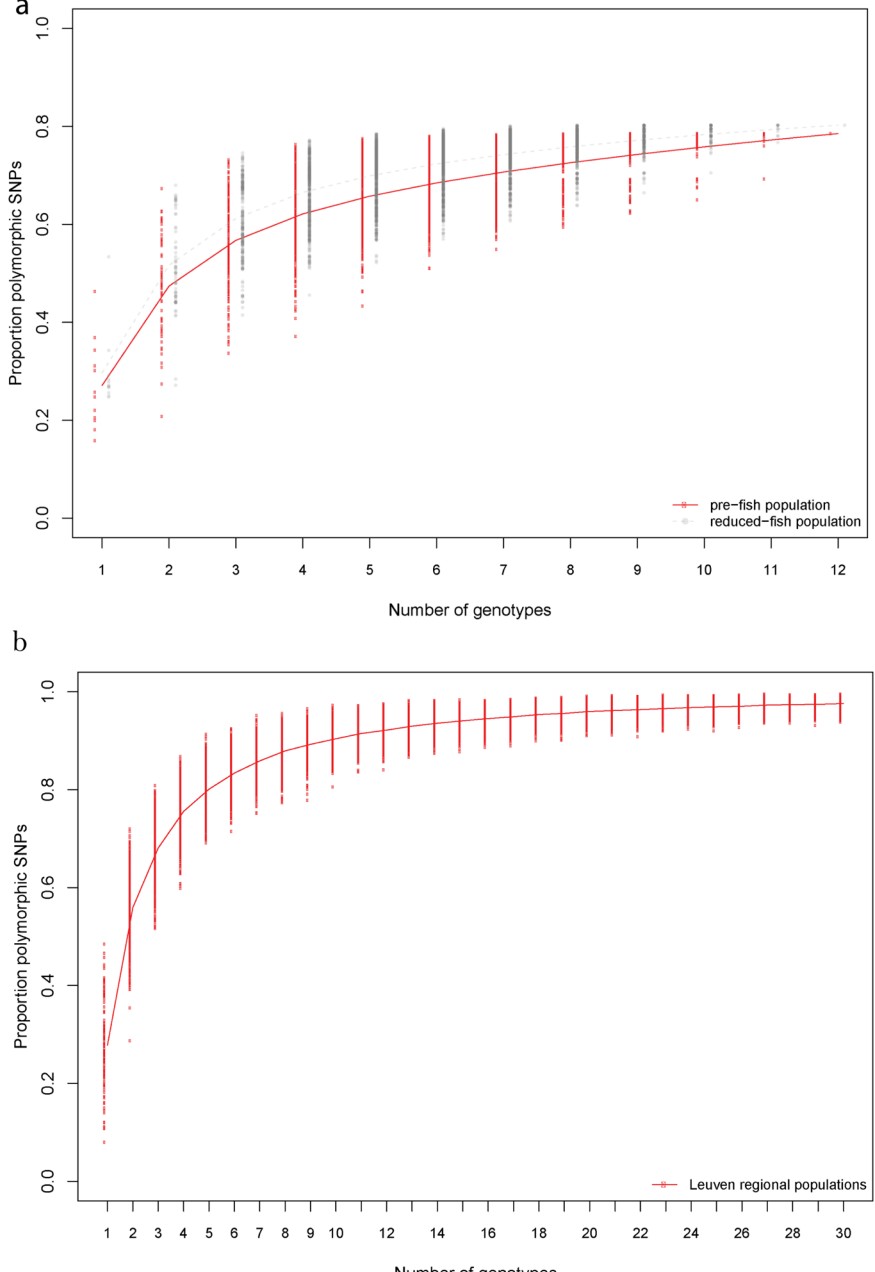

**Fig. 4 Rarefaction curves for the proportion of polymorphic SNPs. a** The proportion of SNPs ($N = 1109$) that were potentially under positive selection (i.e., excluding hitchhiked loci) in the OHZ population that is polymorphic as a function of random samples of an increasing number of individuals from the pre-fish (solid, thick red line) and reduced-fish (hatched, thinner gray line) temporal subpopulations. The lines show that the amount of polymorphisms captured by even a low number of individuals is high in both populations, and that prior exposure to selection results in an even higher amount of polymorphism in the reduced-fish compared with the pre-fish population. **b** The proportion of SNPs that were potentially under positive selection (i.e., excluding hitchhiked loci) in the OHZ population that were also found in the spatial dataset ($N = 1003$; i.e., 90.4% of the SNPs from Fig. 4a) that is polymorphic as a function of an increasing number of individual genotypes from the nine Leuven regional populations (i.e., the populations in the immediate neighborhood of OHZ) (solid, thick red line).

provides a high evolutionary rescue potential[18] permitting effective adaptive responses to environmental change, as observed in other studies on natural *Daphnia* populations[19,20].

In addition to delivering rare insights into the temporal dynamics of standing genetic variation (see also Bergland et al. 2014[21]), our study also yields empirical evidence for the evolutionary rescue potential that even a few colonists derived from multiple populations in the landscape can provide. A modeling study similarly predicted that a few migrants might facilitate rapid freshwater adaptation in sticklebacks[22], and a study on

carabid beetles living in habitats that differ in predictability revealed that as few as ten individuals in most cases were enough to provide the standing genetic variation to allow evolution from long-winged to short-winged morphs[23]. Standing genetic variation seeded during colonization or subsequent immigration by a limited number of genotypes whose genome bears the legacy of divergent selection in the past might be a common facilitator of rapid evolution, and might explain the growing number of studies reporting rapid adaptive evolution in nature[1]. Rapid evolution fueled by standing genetic variation has been shown in both

vertebrate[14,23–25] and invertebrate taxa[26,27]. The insights offered by our study on the origin and maintenance of standing genetic variation are highly relevant in the context of how evolution can modulate ecological responses to global change[28–31].

The emerging pattern from our study is one of high regional genetic polymorphism that is likely maintained at the landscape level through differential selection regimes across ecologically divergent habitats[26], and of high local genetic polymorphism that is derived from regional founders or immigrants and maintained by large population sizes and fluctuating selection[32]. We indeed observed high standing genetic variation in all regional populations studied (Supplementary Fig. 6). This may be crucial for the long-term maintenance of regional diversity, as it ensures that a few colonists or immigrants seed diversity in local populations, irrespective of their origin. The resulting local standing genetic variation allows divergent responses to environmental gradients, which in turn again contributes to the maintenance of regional genetic diversity. Thanks to the large effective population sizes, local populations can be ecologically responsive while maintaining evolutionary potential to respond to future changes.

We suggest that the patterns we observe are likely general, given that many species, especially invertebrates, are often characterized by large effective population sizes[33], and most traits are polygenic. Our results also testify to the importance of standing genetic variation. Maintenance of genetic diversity is an important biodiversity goal[34], and likely requires multiple sufficiently large populations because of the interdependence of local and regional genetic diversity. Protection and restoration of the evolutionary potential of larger-bodied and less common species that are characterized by low effective population sizes and suffer from loss of evolutionary potential[35,36] should be a high priority, and requires a landscape perspective.

## Methods

**Study system**. We studied a *D. magna* population (OHZ) from a small, shallow man-made pond in Oud-Heverlee, Belgium (50°50′ N– 4°39′ E). This pond was constructed for pisciculture in 1970 and has a detailed record of fish-stocking densities for 16 years (Fig. 1a). Dormant stages of *D. magna* were sampled from three depths of a sediment core, corresponding to three time periods that varied in the level of fish-predation pressure: (1) the pre-fish period (1970–1972), during which no fish were stocked in the pond; (2) the high-fish period (1976–1979), a period with high fish-predation pressure due to intensive fish stocking; (3) the reduced-fish period (1988–1990), with relaxed fish predation pressure due to a reduction in fish stocking (Fig. 1a)[9,10,37]. This archive was previously sampled using a standard Plexiglas corer with inner diameter of 5.2 cm[10]. Dating of the sedimentary archive could not be completed with traditional radioisotope analysis, but was based on dry weight and organic matter content under the assumption of a constant sedimentation rate since the establishment of the pond[10]. The cores contained the full sediment archive, including the transition to the mineral sediment. Sediment cores were aligned using the patterns of *Daphnia* dormant egg abundance and changes in size of the dormant egg cases as described in Cousyn et al.[10]. The dormant stages were hatched in the laboratory and taking advantage of the parthenogenetic reproduction mode of *D. magna* as long as conditions are favorable, we started up clonal lines. The resulting clonal lines are each genetically unique, as dormant stages in *D. magna* are the result of sexual reproduction. Our approach was thus to sequence the full genome of a random sample of 12 individuals from each of three depth layers of a sediment core representing populations that occurred in three periods with distinct fish-predation pressure.

In addition to the reconstruction of temporal genome dynamics, we used twelve regional populations of *D. magna* distributed along strong environmental gradients of fish-predation pressure in the region. Six populations (DANA, U2, TER1, MO, KNO15, and TER2) were sampled from fishless ponds, while six populations (ZW4, LRV, ZW3, OHN, OM2, and OM3) were sampled from ponds that harbored fish (Supplementary Table 1). These genotypes were hatched from dormant eggs isolated from the upper 2–3 cm of sediment of the study ponds.

**Whole-genome sequencing**. To reconstruct the genomic history, we resequenced the 36 *D. magna* lines resurrected from the OHZ pond and validated it with additional whole-genome resequencing of 144 *D. magna* genotypes spread across twelve spatial populations along a fish gradient in the region (Supplementary Table 1). Twelve individuals from each temporal subpopulation of the sediment core and 8–17 individuals per population in the spatial survey were used for

genomic DNA extraction using the Nucleo Spin Tissue extraction kit (Macherey-Nagel, Germany), with overnight incubation at 56 °C and following the manufacturer's instructions. We quantified DNA using PicoGreen reagent (Life Technologies) on a DTX 880 spectrofluorometer (Beckman Coulter). For each sample, 1 μg of gDNA was normalized in a final volume of 50 μl of Tris Buffer, pH 8.5, and sheared using an E220 Focused Ultrasonicator in conjunction with a microTube plate (Covaris) in accordance with the manufacturer's recommendations. Sheared genomic DNA was assayed on a 2200 TapeStation (Agilent) with High Sensitivity DNA Screentapes to determine the distribution of sheared fragments. The sheared genomic DNA was then prepared into Illumina compatible DNA Sequencing (DNASeq) 100-bp paired-end libraries utilizing NextFlex chemistry (Bio Scientific). Following library construction, libraries were assayed on a 2200 TapeStation (Agilent) with High Sensitivity DNA Screentapes to determine the final library size. Libraries were quantified using the Illumina Library Quantification Kit (Kapa Biosystems) and normalized to an average concentration of 2 nM prior to pooling. Genomic DNA quantification and normalization, shearing setup, library construction, library quantification, library normalization, and library pooling were performed utilizing a Biomek FXP dual-hybrid automated liquid handler (Beckman Coulter). C-Bot (TruSeq PE Cluster Kit v3, Illumina) was used for cluster generation and the Illumina HiSeq2500 platform (TruSeq SBS Kit v3 reagent kit) for paired-end sequencing with 100-bp read length following the manufacturer's instructions.

**Short-read mapping and variant calling**. The paired end reads (100 × 2) of each individual were first analyzed using FASTQC (http://www.bioinformatics.babraham.ac.uk/projects/fastqc/) for quality checks. Subsequently, low-quality base trimming and adapter cleaning was performed using the Trimmomatic software[38]. Here, parameter values to remove adapter sequences were chosen for seedMismatches (2), palindromeClipThreshold (30), and simpleClipThreshold (10). The minimum phred quality required to keep a base was set to 28, and the minimum read length to 50 bp. Furthermore, the cleaned reads were mapped to the *D. magna* genome version 2.4[39] using Bowtie2[40] software with very-sensitive parameter settings (-D 20 -R 3 -N 0 -L 20 -i S,1,0.50) and insert size between 200 and 700 bp. The mapped reads were then marked for duplicates using the MarkDuplicates feature of Picard tools (http://broadinstitute.github.io/picard/) to avoid PCR duplicates. The resulting sorted BAM files were then used for variant calling using FreeBayes[41]. FreeBayes[41] is a haplotype-based variant caller that calls SNPs, indel, and complex variants. Minimum base quality was set to 30 with minimum coverage of four reads. We obtained more than $3 \times 10^6$ raw variants (3 441 615) for the OHZ temporal subpopulations and $6 \times 10^6$ raw variants for the spatial populations.

Only biallelic SNPs supported by at least four reads and sequenced in at least 90% of individuals were retained after filtering. The draft genome of *Daphnia magna* consists of thousands of scaffolds and contigs. To remove repetitive and paralogous regions in the genome, we used the 293 scaffolds greater than 5 kb that altogether represent 84% of the sequenced genome. Further SNP filtering was performed based on *D. magna* gene models, such that each polymorphic SNP contained within genic regions could be unambiguously assigned to only one gene locus, thereby removing uncertainties attributed to sequence reads mapping to paralogs or to overlapping genes coding on alternative strands of DNA. Finally, SNPs at frequencies below 5% (pooled subpopulations) were removed retaining a total of 724,321 SNPs (mean coverage 20 reads per SNP/individual; 99.6% SNPs with missing values less than 5%) for the temporal analysis of the OHZ population and 748,511 SNPs for the spatial populations. These SNPs were used for downstream analyses.

**Population differentiation**. We calculated both genome-wide and locus-specific levels of genetic differentiation ($F_{ST}$; Weir & Cockerham 1984[42]) using the diffCalc function of the diveRsity[43] package in R[44]. These calculations were performed for each pair of temporal subpopulations (i.e., pairwise $F_{ST}$) in the temporal setting (OHZ) and for six random pairs of nonfish and fish populations in the spatial survey.

To calculate allele frequencies in the temporal analysis, we used vcfglxgt function of vcflib (https://github.com/vcflib/vcflib) to set genotypes that are most likely to be true based on maximum genotype likelihood. We then identified the significant differences in allele frequencies between temporal subpopulations over time using a modified chi-square test developed by RS Waples (Waples 1989)[11] and implemented in the TAFT software[45] (hereafter referred to as Waples test) that accounts for effective population size (Ne), yielding 30,669 significant SNPs (4.23% of total OHZ SNPs) by comparing the prefish and high-fish temporal subpopulation and yielding 11,257 SNPs (1.55% of total OHZ SNPs) for the high-fish and reduced-fish temporal subpopulation comparison; 1771 SNPs showed significant allele-frequency changes in both transitions, most of them also showing a significant reversal in the second transition. To determine whether the observed number of SNPs that showed a significant reversal in allele frequencies is higher than expected by chance, we estimated the null distribution by randomly permuting the temporal subpopulation labels (i.e., prefish, high fish, and reduced fish) and alleles per locus (724,321 SNPs), and recalculating the number of reversals based on Buffalo and Coop 2020[12].

**Estimation of effective population size (Nₑ).** Effective population sizes ($Ne$) were calculated from $\theta = 4Ne\mu$, across the whole genome and with a mutation rate per generation of $4 \times 10^{-9}$[46] and a generation time of one year (*Daphnia* undergoes 10–15 asexual generations and one sexual generation per year), where $\Theta$ is Watterson's diversity index and μ is mutation rate. Watterson $\Theta$ was calculated using the folded SFS option in ANGSD software[47] and found to be stable, i.e., near 0.03 across the three subpopulations (prefish, high fish, and reduced fish). The calculated Ne was found to be ~1.66 million in the prefish temporal subpopulation and ~1.72 million for the high-fish and reduced-fish temporal subpopulations. Similarly, for spatial populations, the value ranges from ~1.06 to 1.45 million (Supplementary Table 2).

**Detecting genomic islands of differentiation.** For each scaffold, and for each pairwise comparison among temporal subpopulations and six independent fish and no-fish replicate pairs in the spatial survey, a hidden Markov model (HMM) was used to distinguish genomic regions of high, moderate, or low differentiation among (sub)populations. We used a similar approach as used earlier by Soria-Carrasco et al. (2014)[48]. In brief, for each of these three levels of genetic differentiation (i.e., the hidden states), a Gaussian distribution of $\log10(F_{ST} + 1)$ was assumed with the mean and variance initialized as those of the $\log10(F_{ST} + 1)$ values within each respective level. We then used the Baum–Welch algorithm[49] to refine the Gaussian model for each state and the transition matrix among the states. Direct transition from the low to the high state was not allowed. Hidden states were then estimated from the data and we estimated the parameters by the Viterbi algorithm using the R package HiddenMarkov[50]. A high differentiating island between genomes is defined to contain at least three consecutive SNPs categorized as high-state SNPs by HMM, yielding 6111 and 2879 islands of genomic differentiation between the prefish vs high-fish (mean length: 2428 bp) and the high-fish vs reduced-fish comparison (mean length: 1713 bp), respectively. Similarly, for six independent spatial fish vs no-fish comparisons, the number of islands of differentiation ranged between 4136 and 7493 (with range of mean length 1879–3290 bp), depending upon the comparison (Supplementary Table 3).

**Functional annotation and enrichment analysis.** We investigated the function of the outlier SNPs ($P$ values smaller than 0.01) in the comparison among temporal subpopulations (prefish vs high fish and high fish vs reduced fish) and in HMM-based high-differentiation islands in spatial comparisons. Transcriptome-based functional annotation was performed using the *Daphnia magna* genome version 2.4[39]. The pathway enrichment analysis was performed using the orthologous genes of *D. magna* in the *D. pulex* genome[51] based on OrthoDB gene families[52] and the KEGG pathway database[53]. Out of ~29,000 annotated genes of *D. magna*, 17,400 genes have 17,832 orthologs in *D. pulex*. However, due to the fragmented status of the *D. magna* genome assembly, manual curation for high-quality gene models resulted in a total of 12,264 *D. magna* genes used in our study, of which 2402 genes are annotated to KEGG pathways. Ortholog mapping is not unique. A given gene from the source species, here *D. magna*, can map to a single, multiple, or no ortholog in the target species, here *D. pulex*. This can bias statistical tests when referencing to *D. pulex* genomics resources. We used the number of nonunique mappings for each *D. magna* gene on the KEGG pathways of *D. pulex* to weight-adjust the confusion matrix for Fisher's exact test to obtain the correct P values. Significant pathways are defined as those with FDR corrected (Benjamini–Hochberg method) P values smaller than 0.05. The data analysis was performed using Python packages (NumPy v1.17.4[54], SciPy v1.4.1[55], statsmodels v0.11.1, and plotly v4.8.1).

**Rarefaction analysis.** Rarefaction analyses were used to determine the rate at which outlier SNPs accumulate in the temporal subpopulation or in the set of regional populations as a function of sample size, i.e., the number of individuals sampled from a given population or group of populations. These analyses were performed separately for the prefish as well as for the reduced-fish temporal subpopulation, in both cases to assess the number of individuals needed to accumulate a given percentage of the SNPs that were suggested to be important for the evolution in response to fish through an outlier analysis of the prefish to high-fish transition. With the rarefaction analysis on the prefish population, we estimate the minimum number of individuals that are needed to reach sufficient genetic variation to enable the observed level of adaptation to fish in this population. The rarefaction analysis on the reduced-fish temporal subpopulation was to assess whether the level of genetic polymorphism declined or increased following the period of strong selection by fish. We thus aimed to evaluate how much evolutionary potential a certain number of individuals from the oldest (i.e., before the introduction of fish) as well as the youngest temporal subpopulation (i.e., after a wave of selection) represent. In the first set of analyses, we used the 1109 SNPs belonging to the divergent SNPs that showed significant changes in allele frequencies in the prefish to high-fish transition and also a significant reversal during the high-fish to reduced-fish and were potentially under positive selection (i.e. excluding hitchhiked SNPs) in the OHZ population. This group of SNPs represent polymorphisms that are presumably adaptive or at least contribute to adaptive allelic variants and hence contribute to the adaptive potential of the temporal

subpopulations. The analyses were performed by rarefying the genotype matrix of all 12 individuals from either the prefish or the reduced-fish population to all possible (i.e., 4095) subsets of samples of 1–12 genotypes. For each of these subsets we then calculated the average proportion of polymorphic SNPs. These values were plotted against sample size to generate rarefaction curves. Similarly, rarefaction analyses were performed for the 1003 SNPs belonging to the outlier SNPs that were potentially under positive selection (i.e., excluding hitchhiked SNPs) in the OHZ population and that were also present as SNPs in the full spatial dataset (90.4% of the total number of 1109 SNPs). In this case, rarefaction curves were plotted by randomly resampling (1000 times) 1–30 individuals from the total of the 111 individuals of the Leuven regional populations in the spatial data set (i.e., the cluster of populations that represents a sample of nearby populations (within a radius of 10 km) to the focal OHZ population).

**Reporting summary**. Further information on research design is available in the Nature Research Reporting Summary linked to this article.

## Data availability

All genome-sequencing data generated in this project are available at NCBI BioProject under accession code PRJNA344883 and PRJNA624267.

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

## Acknowledgements

We gratefully acknowledge financial support from KU Leuven Research Fund projects PF/2010/07 and C16/2017/002, from FWO projects G.0468.10, G.0614.11, G0B9818, and G0C3818, and from NERC project NE/N016777/1. KIS was supported by a PhD fellowship of the Agency for Innovation by Science and Technology in Flanders (IWT) and TC was supported by a DFG Research Fellowship (CZ 230 1/1). We thank Edson Sandoval-Castellanos for providing source code of the TAFT software and the *Daphnia* Genomics Consortium, including Don Gilbert (Indiana University) for its collective efforts to assemble and annotate the *D. magna* genome.

## Author contributions

A.C., L.O., K.I.S., and L.D.M. conceived the study; L.O. coordinated the full genome sequencing effort; A.C., J.Z., J.A.M.R., T.C., C.E.J., and K.I.S. performed the data analysis with input from J.K.C. and L.D.M.; A.C., J.Z., L.D.M., J.K.C., and J.A.M.R. wrote the paper with input from T.C., J.R.S. and L.O. All authors discussed the results and commented on the different versions of the paper.

## Competing interests

The authors declare no competing interests.
