## [Peer Review File · Nature Communications]

REVIEWER COMMENTS

Reviewer #1 (Remarks to the Author):

Chaturvedi et al present WGS data from *Daphnia magna* populations experiencing varying levels of fish predation and fish-induced selection. I really liked reading this paper and have almost nothing "negative" to say about it (in fact, this might be the shortest review I ever wrote). The methods and analytical approaches all seem solid, although someone with more experience analyzing WGS data would be in a better position to comment. I only have a few minor comments, mostly about packaging.

Title: very broad. The study does have implications for the origin, maintenance, and impact of standing genetic variation, but none of these things are studied directly. For example, several mechanisms responsible for the origin and maintenance of genetic variation (spatially-divergent selection, dispersal) in these populations are discussed but not measured explicitly. I would use a more specific title, e.g. "Extensive standing genetic variation enables rapid adaptation in *Daphnia magna*" (just a suggestion, not necessarily this title..)

27: 'Whole' genome sequencing instead of just genome sequencing?

33: I would end the abstract with a 'take-home message' sentence instead of one (of several) key finding.

56: I would close the intro by stating your hypotheses clearly. E.g., "we predict that: 1) ... , 2) ... "

82: Is the first transition 'stronger' because it also involves adapting to a novel environment? i.e., individuals were collected shortly after the pond was colonized by *Daphnia*, so could some of the allele frequency change be due the population adapting to this new human-made pond? Is it possible that the alleles that reverse in trend upon relaxation of fish predation are those involved in adaption to fish, whereas those alleles that keep increasing in frequency even after fish predation diminishes might be alleles involved in adaptation to this novel pond environment?

84: Are these values (8 and 4 % of the genome) that small? It would be helpful to come back to these numbers in the discussion and compare your results with those from other study systems. Based on other studies, what fraction of the genome would be expected to change in response to this selection pressure?

133-137: why specify and focus on this one particular pathway? (dpx04080). Other pathways respond even more. I would list the 3 main pathways that responded the most (dpx00230, dpx00983, and

dpx04080) and provide results of statistical test for each.

134: here and elsewhere: Fig. SX not fig.

150: Perhaps list exceptions (pathways that respond differently in the spatial and temporal surveys). For example Fig. S3 made me wonder what was happening with dpx00601 and dpx00603

169-172: Is this also true in the spatial survey, i.e. in ponds where fish have been present for longer than in OHZ? You mention in the discussion that fluctuating selection + large N_e = high levels of polymorphism, but it would be nice to show that there's no fixation of alleles even in ponds with a longer exposure to fish. Could you add pond-specific lines to Fig 4B underneath the landscape-level curve that you are showing?

180: "nine inland populations": this is difficult to understand without seeing the map in SM. Perhaps introduce earlier that 3 of the NF ponds were close to the focal pond with resurrected populations, and 3 were away near the coast. In fact, why include these 3 distant ponds? Do you need 6 'pairs' of sites, or couldn't you simply compute all pairwise comparisons between the 6 F ponds and the 3 nearby NF ponds? What is the advantage of including these 3 distant sites that might not be comparable in terms of water chemistry, etc.?

189-209: The discussion does not include a single reference! I definitely think that some comparison with other study systems would be beneficial. This section is also very brief and a little speculative (eg the section about spatially divergent selection and dispersal across the landscape). Can you strengthen your argument by citing other studies that showed the importance of these processes for the maintenance of standing genetic variation? Can you say a few things about the conservation implications of your results? I think your results suggest that populations are very unlikely to be limited by standing genetic variation when adapting to global change. This has profound implications and could be tied back to the literature on human-induced evolution, the importance of genetic diversity in conservation, etc. I had difficulties grasping the main take-home message from your study and see how your results compare with what has been found in other systems.

226: "to fully genome sequence a random sample" -> "to sequence the full genome of a random sample"

228: How was the core dated? I imagine this information is provided in earlier papers but perhaps a few words on the methods would help.

229-233 & 237-239: repetitive.

319-327: it is not clear to me whether you had an a priori hypothesis about N_e . The results on N_e come a bit out of the blue.

339: disallowed -> not allowed.

394: missing a closing parenthesis

483: "this population underwent rapid (...) adaptive evolution" instead of "underwent a rapid and adaptive evolution"

Fig 4B: again, hard to understand what 'inland' means without consulting the SM

Reviewer #2 (Remarks to the Author):

Sequencing *Daphnia magna* populations that vary in fish predation, through time and across space, the authors claim to have identified abundant standing genetic variation that responds rapidly, and adaptively, to selection. Documenting the temporal dynamics of rapid polygenic adaptation at the genome level outside the laboratory is relatively novel and of general interest in evolutionary biology. The claim that the variation needed for rapid evolution is ubiquitous and not eroded by selection will, I think, influence many readers. However, I remain unconvinced for the following reasons:

1) On lines 154-156 the authors claim to have evidence that selection has acted on standing genetic variation rather than new mutation (a major claim of the paper more generally). I am unsure. First, they seem to have excluded SNPs at frequencies below 0.05 (clarification needed: is this for all populations pooled or each population? line 277), which may have removed many large effect mutations and thus biased the results towards standing variation. Second, given that the authors observed many islands of differentiation (blocks of the genome with high F_{st}), hitchhiking with new (or rare) mutations seems a very plausible alternative, as common standing variants will have much less linkage disequilibrium with surrounding SNPs (or are these islands too small to be consistent with the LD created by new mutation?).

2) I do not buy the argument that all genes in overlapping islands of differentiation (ie differentiated in both the first and second temporal transition) are under positive selection (line 127). There is very likely a ton of linkage disequilibrium (and thus hitchhiking) in both transitions, and so I would guess the authors still greatly overestimate the number of genes involved here.

3) More fundamentally, I am unsure about the significance test used to determine what SNPs are "outliers" (line 78). Why is the F-distribution used? How were its parameters chosen? Was it fit? How good was the fit? What null model does it test? Does it include drift (N_e) and the relatedness of two populations (genome wide F_{st})? I am similarly confused about the Chi-squared test to test allele frequency differences (line 95). These tests are the basis of all claims and so need to be clearly justified.

4) Finally, the connection between the SNPs and phenotypic adaptation is still very much unclear,

despite the authors claims (lines 159-161). Where is the evidence that these genetic changes were adaptive, or even connected to the phenotype? Given abundant genetic variation and clonal reproduction it is presumably possible to perform common garden experiments or GWAS? Are the traits also quite variable?

Minor comments:

line 46-47: selective sweeps are actually the reason we can often find the loci targeted by selection

line 47-48: I am not sure why a failure to predict if populations evolve in reverse or adapt to other things is the fault of the sweep

line 81: given N_e and the number of generations that has passed you should be able to predict the expected F_{st} due to drift alone

line 84: I am not sure nearly 10% of the genome is a "small fraction"

line 91-98: could you perform permutation tests to statistically test whether the number of reversals is significant? (see Section S6 of Buffalo & Coop 2020 bioRxiv 798595)

lines 105 and 108: I find it clearer to differentiate sweeps (increase in frequency due to positive selection) from hitchhiking (increase in frequency due to linkage with a sweeping allele)

line 118: aren't the islands of divergence in the second transition measured relative to the high fish population, and therefore only 10 years has elapsed (rather than 16)?

line 179: a rarefaction curve within a population would be interesting too, as it would say whether the variation is already there locally, without the need for migration. I guess the low F_{st} suggests all the variation is indeed already in each population

line 199: again confused about SNP filtering -- you would include SNPs that fixed?

line 280: so you only include SNPs in genes? this seems like a major understated decision as the paper discusses the genome, the percent of the genome under selection, etc

line 313-314: unclear -- these SNPs changed allele frequency in both transitions?

line 336: Gaussian $\log_{10}(F_{st})$ seems like an odd choice given F_{st} can be zero (what did you do then?) and is maximized at 1 (while the Gaussian goes to infinity). I see Soria-Carrasco et al (cited) use a logit, which does not have these problems. Why the change?

line 336-337: how can you initialize like this before you know which SNPs are in which level?

line 338: was the same Gaussian model (ie the mean and variance of $\log_{10}(F_{st})$ within each level) used for both temporal transitions? If not, is it fair to compare levels of differentiation across the transitions?

line 340: "estimated parameters" has some grammar issues, maybe "we estimated the parameters"

line 369: in -> as a

line 377: genetic -> adaptive genetic (otherwise theta tells you the answer)

line 380: say where 2204 comes from

line 394: missing a ")"

line 483: underwent a rapid -> underwent rapid

fig 2: why do some alleles start below 0.05 (despite filtering)? Do I misunderstand the filtering or is this a result of the binomial model of frequency you use?

Supplementary material:

line 66: "pair" -> "triplet"

line 74: how can you see these three categories in the plot? It seems like only "high" is coloured? (In general it is not clear why the moderate class exists.)

REVIEWER COMMENTS

Reviewer #1 (Remarks to the Author):

Chaturvedi et al present WGS data from *Daphnia magna* populations experiencing varying levels of fish predation and fish-induced selection. I really liked reading this paper and have almost nothing "negative" to say about it (in fact, this might be the shortest review I ever wrote). The methods and analytical approaches all seem solid, although someone with more experience analyzing WGS data would be in a better position to comment. I only have a few minor comments, mostly about packaging.

A1. We thank the reviewer for her/his positive assessment of our manuscript.

Title: very broad. The study does have implications for the origin, maintenance, and impact of standing genetic variation, but none of these things are studied directly. For example, several mechanisms responsible for the origin and maintenance of genetic variation (spatially-divergent selection, dispersal) in these populations are discussed but not measured explicitly. I would use a more specific title, e.g. "Extensive standing genetic variation enables rapid adaptation in *Daphnia magna*" (just a suggestion, not necessarily this title..)

*A2. We changed the title and it now reads "Extensive standing genetic variation from a small number of founders enables rapid adaptation in *Daphnia*"*

27: 'Whole' genome sequencing instead of just genome sequencing?

A3. We agree and changed it into "whole genome sequencing"

33: I would end the abstract with a 'take-home message' sentence instead of one (of several) key finding.

A4. We edited the abstract to add the following take home message: "Our results provide insight on how natural populations can acquire the substantial genomic variation, through colonization by a few regional genotypes, that fuels rapid evolution in response to strong selection pressures, which can involve hundreds of genes without eroding genomic variation." This required reducing the wording of other parts of the abstract.

56: I would close the intro by stating your hypotheses clearly. E.g., "we predict that: 1) ... , 2) ... "

*A5. We have now added a paragraph stating four specific hypotheses at the end of the Introduction section: "Resequencing the genomes of resurrected genotypes of a natural *Daphnia* population that shows an elaborate evolutionary response to changes in fish predation pressure, we here test the hypotheses that (i) the rapid evolutionary response observed in this population is largely based on standing genetic variation and that (ii) this standing genetic variation is maintained through time irrespective of the responses to selection. By also sequencing genotypes from a set of populations from contrasting habitats in terms of fish predation pressure, we also test the hypotheses that (iii) standing local genetic variation reflects regional standing genetic variation and that (iv) a few founding individuals from the regional genotype pool suffice to establish the extensive standing genetic variation fueling rapid evolution in natural populations"*

82: Is the first transition 'stronger' because it also involves adapting to a novel environment? i.e., individuals were collected shortly after the pond was colonized by Daphnia, so could some of the allele frequency change be due the population adapting to this new human-made pond? Is it possible that the alleles that reverse in trend upon relaxation of fish predation are those involved in adaption to fish, whereas those alleles that keep increasing in frequency even after fish predation diminishes might be alleles involved in adaptation to this novel pond environment?

A6. While in principle this could be the case, it is quite unlikely, as the traits involved in the adaptation (studied by Stoks et al 2016) all shift from a state that is adapted to a non-fish condition to a state that is adapted to high fish predation pressure. If anything, the first three years of the pond's existence likely allowed the local population already to adapt to local conditions (e.g. the absence of fish). The difference in strength of the response likely reflects that in the first transition the change in environment was very strong and "qualitative" (from no fish to high fish densities) whereas in the second transition the reversal in selection conditions was only partial (from high fish to reduced fish; there were still quite some fish present so it is not a reversal to a non-fish condition).

84: Are these values (8 and 4 % of the genome) that small? It would be helpful to come back to these numbers in the discussion and compare your results with those from other study systems. Based on other studies, what fraction of the genome would be expected to change in response to this selection pressure?

A7. This is indeed a matter of perspective and scale. We refer to the impact to be rather modest because less than 10% of the SNPs and less than 1% of the genome is influenced. Translated into the number of SNPs or genes that is affected, it still amounts to quite an impact - almost 500 genes are directly impacted (which is still a low percentage of the total number of genes). So, it depends on whether one looks at the percentage or at the sheer number of sites/genes involved. One reason to call it "modest" is that in our study on >10 life history and behavioural traits (Stoks et al 2016), all but one trait showed a significant evolutionary response. We implemented the suggestion of the reviewer in the discussion of the manuscript. See lines L259-267.

133-137: why specify and focus on this one particular pathway? (dpx04080). Other pathways respond even more. I would list the 3 main pathways that responded the most (dpx00230, dpx00983, and dpx04080) and provide results of statistical test for each.

A8. We implemented the suggestion of the reviewer in the manuscript. See lines L174-179.

34: here and elsewhere: Fig. SX not fig.

A9. Implemented

150: Perhaps list exceptions (pathways that respond differently in the spatial and temporal surveys). For example Fig. S3 made me wonder what was happening with dpx00601 and dpx00603

A10. Overall, the patterns generated by the comparisons among populations (spatial analysis) are less easy to interpret than the one in the temporal analysis, as the temporal analysis captures a clear-cut change in time within a given population, i.e. a given genetic background. In space we contrast populations that differ in fish predation pressure but they may also differ in other aspects, and this may differ among pond pairs. This is why it is important to go for patterns that are repeated across more than

one comparison. Focusing on the exceptions is therefore not necessarily useful, because it may be that two or three pairs differ not only in fish but, e.g., also in eutrophication level or parasites.

169-172: Is this also true in the spatial survey, i.e. in ponds where fish have been present for longer than in OHZ? You mention in the discussion that fluctuating selection + large N_e = high levels of polymorphism, but it would be nice to show that there's no fixation of alleles even in ponds with a longer exposure to fish. Could you add pond-specific lines to Fig 4B underneath the landscape-level curve that you are showing?

A11. The referee is very right. Most of the 1003 SNPs relevant for the response to selection in the temporal population are also not fixed in the spatial populations, as illustrated by the plot below (% Polymorphic SNPs per spatial population). Given that we do not have strong information on the degree to which the fish predation pressure was stable in all of these populations, however, we prefer not to emphasize this, to avoid overinterpretation of the results of the spatial analysis. But we now show the plot in SI (Fig S6) and below (Fig R1).

Fig R1. The proportion of SNPs that are polymorphic in each pond from the 12 sampled spatial populations (i.e. six populations from ponds without fish (NF) and six populations from ponds with fish (F)) and were under positive selection (i.e. excluding hitchhiked loci) in the OHZ population and that were found in the spatial dataset ($N = 1003$; i.e. 90.4% of the loci from Fig. 4A)

180: "nine inland populations": this is difficult to understand without seeing the map in SM. Perhaps introduce earlier that 3 of the NF ponds were close to the focal pond with resurrected populations, and 3 were away near the coast. In fact, why include these 3 distant ponds? Do you need 6 'pairs' of sites, or couldn't you simply compute all pairwise comparisons between the 6 F ponds and the 3 nearby NF ponds? What is the advantage of including these 3 distant sites that might not be comparable in terms of water chemistry, etc.?

A12. We understand the comment of the reviewer, as it would make the "regional nature" more clearly defined (ie. 10 km around the target pond). We carried out this analysis by removing the 3 coastal populations, and the results are in essence similar to the results with those populations included. See figure below Fig. R2 and Fig. S7. We do, however, prefer to keep the design of the analysis as we performed it for the original submission for a statistical reason: if we only use the data from the inland ponds, then we need to use the data of some non-fish ponds more than once (as there are more fish than non-fish ponds), and this creates non-independence that would make our statistical analysis much less trust-worthy. In addition, the pattern would then in essence be dependent on three Non-fish populations, and that provides a less strong basis for conclusions than if we use six independent contrasts, even if those contrasts span a somewhat larger distance.

Fig. R2. The proportion of SNPs that were under positive selection (i.e. excluding hitchhiked loci) in the OHZ population and that were found in the spatial dataset ($N = 1003$; i.e. 90.4 % of the loci from Fig. 4A) that is polymorphic as a function of an increasing number of individual genotypes from the 12 sampled spatial populations (i.e. six populations from ponds without fish (NF) and six populations from ponds with fish (F)) (solid, thick red line).

189-209: The discussion does not include a single reference! I definitely think that some comparison with other study systems would be beneficial. This section is also very brief and a little speculative (eg the section about spatially divergent selection and dispersal across the landscape). Can you strengthen your argument by citing other studies that showed the importance of these processes for the maintenance of standing genetic variation? Can you say a few things about the conservation implications of your results? I think your results suggest that populations are very unlikely to be limited by standing genetic variation when adapting to global change. This has profound implications and could be tied back to the literature on human-induced evolution, the importance of genetic diversity in conservation, etc. I had difficulties grasping the main take-home message from your study and see how your results compare with what has been found in other systems.

A13. We thank the reviewer to point out these shortcomings in our discussion. We agree and have considerably revised our discussion to include the different aspects asked for by the reviewer. We wanted to keep our discussion as concise as possible, but now realize that more context is needed. In the revised version of the discussion we now:

(1) Compare our results on the percentage of genome and number of genes affected by selection with other studies. Strikingly, our results are of a similar order of magnitude as some other studies, which, however, described adaptation to major changes in environment (eg from marine to freshwater, change in host plant). Here we show similar changes for an environmental variable (fish predation pressure) that can both within and among habitats change fast. From that perspective, the changes we report are massive.

(2) We developed somewhat more the arguments for our scenario on how standing genetic variation originates from colonization or immigration of a few individuals from regional populations that are subject to spatially divergent selection. In doing so, we cite other studies that showed the importance of these processes for the maintenance of standing genetic variation.

(3) We added more content on the implications of our results for (a) eco-evolutionary dynamics, (b) responses to global change, and (c) conservation. Our results suggest that many populations, at least those of common species that have large population sizes, might indeed often not be limited by standing genetic variation when adapting to global change. But this holds only for species that indeed have healthy metapopulations consisting of sufficiently large local populations. This indeed has relevance both for human-induced evolution and the importance of genetic diversity in conservation.

226: "to fully genome sequence a random sample" -> "to sequence the full genome of a random sample"

A14. Implemented

228: How was the core dated? I imagine this information is provided in earlier papers but perhaps a few words on the methods would help.

A15. We have now implemented this information in the methods section (indeed following Cousyn et al 2001).

229-233 & 237-239: repetitive.

A16. We strongly edited the discussion text and changed this to reduce repetition.

319-327: it is not clear to me whether you had an a priori hypothesis about N_e . The results on N_e come a bit out of the blue.

A17. We calculated N_e because it is relevant to explain the maintenance of genetic variance. For cyclical parthenogenetic populations such as water fleas, in the past it was assumed that their N_e would be relatively small because of the clonal structure. The situation is, however, more complex because of the (i) very large number of dormant eggs, (ii) at which stage there is sexual recombination, and (iii) the large number of individuals implies that in most cases there are still many clones coexisting and each clone is represented by many individuals (i.e. reducing the probability of extinction of a clone). So while in the past it was often assumed that effective population size of water flea populations would be small, work in the last decade points to large effective population sizes (Lynch et al. GENETICS 206 (1) 315-332), also reflected by the fact that indeed *Daphnia* populations seem to harbour large amounts of genetic

variation. So, it is difficult to formulate a hypothesis on this, because that hypothesis should be very well contextualized. Our full genome study now provides an important high quality proof that effective population sizes of water flea populations are very large. Key take home messages of our study are that genetic variation can be obtained from just a few colonists / immigrants, and that this genetic variation is not rapidly eroded by selection. Both can be linked to high effective population sizes: as large N_e leads to the maintenance of genetic variation, this does not only lead to high genetic variation locally but also regionally, so that it is more likely that a few immigrants can already yield substantial genetic variation to a local population as it assembles.

339: disallowed -> not allowed.

A18. Implemented

394: missing a closing parenthesis

A19. Implemented

483: "this population underwent rapid (...) adaptive evolution" instead of "underwent a rapid and adaptive evolution"

A20. Implemented

Fig 4B: again, hard to understand what 'inland' means without consulting the SM

A21. We now changed "inland" to "Leuven regional populations" in the revised version of the manuscript to increase clarity.

Reviewer #2 (Remarks to the Author):

Sequencing *Daphnia magna* populations that vary in fish predation, through time and across space, the authors claim to have identified abundant standing genetic variation that responds rapidly, and adaptively, to selection. Documenting the temporal dynamics of rapid polygenic adaptation at the genome level outside the laboratory is relatively novel and of general interest in evolutionary biology. The claim that the variation needed for rapid evolution is ubiquitous and not eroded by selection will, I think, influence many readers.

A22. We thank the reviewer for her/his positive assessment of the novelty of our work and its potential impact.

However, I remain unconvinced for the following reasons:

A23. We hope that with the answers provided below and the edits made in the revised version of the manuscript, we can convince the reviewer and the reader that our data provide a strong case for our key conclusions, being (i) that natural populations can harbour substantial amounts of standing genetic variation that fuels rapid evolution, (ii) that this genetic variation can be obtained through the colonization / immigration of only few individuals, and that (iii) a phase of strong selection impacting several hundreds of genes does not lead to genetic erosion. Please see below for a detailed answer to the questions and comments.

1) On lines 154-156 the authors claim to have evidence that selection has acted on standing genetic variation rather than new mutation (a major claim of the paper more generally). I am unsure. First, they seem to have excluded SNPs at frequencies below 0.05 (clarification needed: is this for all populations pooled or each population? line 277), which may have removed many large effect mutations and thus biased the results towards standing variation.

A24. Filtering alleles below 0.05 threshold is common practice to remove sequencing errors and was here applied to the entire dataset prior to downstream analysis. The removed SNPs do not contain variants with large effects that would have been selected for because the variants that are filtered remained at low frequencies in all three temporal subpopulations without significant change in allele frequency over time.

*Because when checking our data and analyses again we found that 40 000 SNPs out of the total of 626 953 SNPs in the previous version of the manuscript were genotyped with more than 90 % missing data, we have now re-analysed the allele frequencies and all downstream analyses of the whole temporal dataset. To correctly assess the degree of missing data we developed a new script and performed the filtering from scratch using 3×10^6 raw temporal SNPs. In the temporal data-set, the number of SNPs before and after filtering allele frequencies, i.e. removing all alleles at a frequency of less than 0.05, are 919393 and 724321, respectively. The 195072 SNPs that were removed in the temporal dataset do not contain variants with large effects that would have been selected for, because none of these SNPs showed a significant change in allele frequencies neither in the transition from no fish to high fish and in the transition from high fish to reduced fish (based on the Waples test; see Waples 1989, "Temporal variation in allele frequencies: testing the right hypothesis", *Evolution*, 43(6), pp. 1236-1251). (see Fig R3 below)*

Fig R3. A) Allele frequency differences between pre-fish vs high-fish for 195072 SNPs that were removed after filtering allele frequencies below 0.05 in pooled subpopulations. By far most alleles changed less than 0.06 in allele frequency in any of the two transitions. B) $\log_{10}(\text{Waples } p\text{-value})$ distribution between pre-fish vs high-fish comparison for 195072 SNPs that were removed after filtering allele frequencies below 0.05. The continuous line at -2 represents the $p\text{-value cut-off} = 0.01$; the dashed line at -1.30 represents the $p\text{-value cut-off} = 0.05$. None of the allele frequencies in the 195072 SNPs exceeded these thresholds. C and D: same as A and B, respectively, but for the high-fish to reduced-fish transition.

More conceptually, we believe our data convincingly show that the observed evolution is dominantly driven by standing genetic variation. We do not deny that mutations have occurred and we do not claim that we can show that not a single mutation was involved. Our key argument is that most of the evolution as observed in traits (Stoks et al. 2016) is driven by standing genetic variation. The evidence for that is strong and consists of several arguments (both pro standing genetic variation and against a big role of new mutations):

(i) For the hundreds of genes and thousands of SNPs that show significant reversible changes in allele frequencies, most of the variation (80%) is already present in the pre-fish population, i.e. before the selection pressure (fish stocking) was present. Given that our sample consisted of 12 individuals (24 genomes), this is truly striking - as we here sample a population that likely has several hundred millions of dormant eggs. Many not so abundant (less than 4%) allelic variants were no doubt missed. Yet, most of the outlier SNPs were detected in that sample of 24 genomes.

(ii) The interpretation that this is standing genetic variation is further supported by the fact that this variation is not only present in the local population, but also well distributed in the regional gene pool. Just 5-10 individuals randomly sampled from regional populations would bring most of the variation in

an assembling population. This shows how easy it is to accumulate this high level of standing genetic variation.

(iii) If a mutation would occur during any of the two transition periods that would play an important role in the adaptation process, one would expect that this allelic variant should reach relatively high frequencies if it would be the key driver of selection. We show above that none of these more rare variants showed significant changes in allele frequencies. Moreover, any allele that reaches a frequency of 15% in any of the populations, even if just one population (e.g. high fish), would not be filtered out.

Given the high level of standing genetic variation that shows allele frequency changes in line with the reversible adaptive trait change, and given that this variation is also present in potential source populations, rapid evolution driven by standing genetic variation is a much more likely explanation than mutation-driven evolution. It is indeed a striking observation, which we believe might indeed influence how people look at standing genetic variation and rapid evolution. But we believe it might be quite common. It only requires (i) species with large effective population sizes, (ii) allelic variants that have fitness consequences but not to the extent that there is immediate / rapid fixation, (iii) temporal and spatial environmental variation. If these three conditions are fulfilled, one can expect landscapes in which populations tend to locally genetically adapt through allele frequency changes building on a large pool of standing genetic variation that is maintained locally and regionally and that is regularly reshuffled when new populations are colonized or when there is immigration. Adaptive genetic variation can in this way be recycled for long times. This bears some similarities to the re-use of freshwater adaptation genes in stickleback that are polymorphic in marine populations (Jones et al, 2012 Nature 484(7392):55-61) and to the rapid adaptation fueled by standing genetic variation in Pogonus beetles (Van Belleghem et al, 2017 PLoS Genet 14(11): e1007796.). In the latter case, the genetic variants that are involved in rapid local adaptation in dispersal traits following recolonizing Europe after the last ice age pre-date that ice age and likely originated during earlier interglacial periods. In the Daphnia populations we studied, one could imagine that every new population builds up substantial standing genetic variation through 5 to 10 early immigrants that come from different ponds in the region. This standing genetic variation can fuel rapid adaptation. The resulting differences in allele frequencies in the different populations then contribute to the standing genetic variation of another novel population when that population is colonized by immigrants coming from different source populations.

Our results are to some extent striking because they do not easily fit with basic population genetic models and are at odds with several studies focusing on conservation genetics. With respect to the models, we believe that the key reason of the disparity is that most population genetic models do not start with seeding new landscapes with genetic variation coming from elsewhere. In the real world, every immigrant brings in new genetic variations, and if a new region is colonized the immigrants might immediately bring in substantial genetic variation derived from multiple other populations from already occupied regions. The disparity with conservation genetics in our opinion reflects the fact that in conservation genetics there is a, entirely justified, bias towards species with small effective population sizes. Our study involves a species with large population sizes. We argue that these might be common, given the vast populations that many small organisms have. Our results show that these populations might show strong capacity for rapid evolution without loss of genetic variation. The fact that there is a rapidly growing literature showing rapid evolutionary changes in a broad range of organisms suggests that indeed these conditions might be common. Our study shows how such rapid evolution is possible.

Note that we do not claim that there are no barriers for evolution. The Daphnia population studied showed extensive adaptive evolutionary change, but we do not claim that this evolution was not limited by barriers of lack of adaptive genetic variation or by genetic covariation. We document the evolution we

observed and the extensive and reversible changes in frequencies of allelic variants we observe, but we do not claim that in the absence of any barriers not even more fine-tuned adaptive evolution would have occurred.

Second, given that the authors observed many islands of differentiation (blocks of the genome with high F_{st}), hitchhiking with new (or rare) mutations seems a very plausible alternative, as common standing variants will have much less linkage disequilibrium with surrounding SNPs (or are these islands too small to be consistent with the LD created by new mutation?).

A25. We understand where the reasoning of the reviewer comes from, but the point is that in our study the SNPs that are on these islands of differentiation are already polymorphic in the source population - so the islands cannot have formed around new mutations but reflect changes in allele frequencies of alleles that were under selection and some hitchhiking. The fact that the same polymorphic SNPs are also found in other populations in the spatial survey argues against rare mutations of large effect, but rather shows that we detected the signal of evolution on standing genetic variation.

2) I do not buy the argument that all genes in overlapping islands of differentiation (ie differentiated in both the first and second temporal transition) are under positive selection (line 127). There is very likely a ton of linkage disequilibrium (and thus hitchhiking) in both transitions, and so I would guess the authors still greatly overestimate the number of genes involved here.

A26. The reviewer is right that we need to formulate this more prudently. In the revised version of the manuscript, the sentence has been changed, stating that the overlapping regions in the HMM islands harboring significant reversible variation is "putatively" under selection. However, in these overlapping regions there was often only one gene. So, while several SNPs of a single gene may then still be hitchhiking, the gene involved likely still was under selection (hence causing the HMM island). Out of 436 overlapping regions of the HMM islands harbouring significant reversal loci, 273 contained a single gene. There were 34 overlapping regions with two genes inside, 12 regions with three genes, three regions with four genes, two regions with 5 genes, one region with six genes and one region with seven genes. There were 110 overlapping regions with no genes. While we agree with the reviewer that our analysis is not detecting all hitchhiking SNPs, the core of our results remains unchanged: there are likely several hundreds of genes involved, and the response is fueled by standing genetic variation.

3) More fundamentally, I am unsure about the significance test used to determine what SNPs are "outliers" (line 78). Why is the F-distribution used? How were its parameters chosen? Was it fit? How good was the fit? What null model does it test? Does it include drift (N_e) and the relatedness of two populations (genome wide F_{st})?

I am similarly confused about the Chi-squared test to test allele frequency differences (line 95). These tests are the basis of all claims and so need to be clearly justified.

A27. In the previously submitted version of the manuscript we used the F-distribution to identify outliers. The F-distribution used was derived from the F_{st} within the temporal data set. The F_{st} obtained from Pre-fish vs High-fish and High-fish vs Reduced-fish populations were estimated as an approximately linear function of a F-distributed random variable (denoted as F_1 below). The outliers were defined as SNPs with p -values < 0.05 in an upper-tailed F-test. The process by which we identified outliers using the F-distribution is described below:

The pairwise F_{st} values were estimated following (Weir and Cockerham, 1984):

$$F_{st} = \frac{MSP - MSI}{MSP + (n_c - 1)MSI + n_cMSG}$$

where MSG measures the allele dispersion within each individual. Therefore we have:

$$F_{st} = \frac{F_1 - 1}{F_1 + (n_c - 1) + n_cF_2}$$

with $F_1 = \frac{MSP}{MSI}$, $F_2 = \frac{MSG}{MSI}$ and $MSI \neq 0$, and thus:

$$F_1 = \frac{n_c(F_2 + 1)}{1/F_{st} - 1} + 1 = \frac{2n_cF_{st}}{(1 - F_{st})(1 + F_{is})}$$

where F_{is} is the inbreeding coefficient of an individual with respect to the local subpopulation.

As in our data F_{is} is small (close to 0), we can approximate $F_1 \approx \frac{2n_cF_{st}}{1 - F_{st}}$, with F_1 following an F -distribution $F_1 \sim F(K - 1, n_1 + n_2 - K)$, in which $K = 2$ is the number of populations and n_1 and n_2 are the number of individuals in the two populations, respectively.

Therefore, we modelled the F_{st} as a monotonically increasing and approximately linear function ($R^2=0.99$) of the random variable F_1 . The outliers were SNPs with F_{st} larger than the upper-tail value estimated from F_1 p -value < 0.05 .

Inspired by the reviewer's comments, we now in the revised manuscript use a modified chi-square test (Waples 1989) to study significant changes in allele frequencies, as this test was developed to quantify temporal changes in allele frequencies. This method accounts for the effect of the effective population size (N_e) on allele frequency changes and number of generations over time, providing a more robust approach than the one we used in the original manuscript. In our revised analysis, we use cut-off values of $p \leq 0.01$, which are more conservative than the ones used in our previous analysis. The results of the outlier test conducted with the two methods (previously chi-square and currently Waples test) generate very similar results and, therefore, our main conclusions remain unchanged (Fig R4). This is because at high N_e , both tests become equivalent.

Fig R4. The relationship between the Chi-square test p-value and Waples test p-value in pre-fish vs high-fish ($R^2 = 0.99$) and high-fish vs reduced fish ($R^2 \sim 1$) comparisons. The Pearson correlation coefficients showed a very strong positive linear relationship, indicating that the two outlier significance tests yield similar results.

4) Finally, the connection between the SNPs and phenotypic adaptation is still very much unclear, despite the authors claims (lines 159-161). Where is the evidence that these genetic changes were adaptive, or even connected to the phenotype? Given abundant genetic variation and clonal reproduction it is presumably possible to perform common garden experiments or GWAS? Are the traits also quite variable?

A28. For the same population as studied here in the temporal analysis, and actually for the same set of clones, evolution in a total of 14 different life history and behavioural traits was quantified in a common garden experiment. These data are published (Stoks et al 2016, Ecology Letters). The results of that study are summarized in Figure 1B as mean changes in trait values of the three subpopulations separated in time. This figure emphasizes the reversal in mean trait values, but Stoks et al. (2016) shows and analyses trait values for all clones. In this study, trait values were quantified both in the absence and presence of fish kairomones (medium conditioned by the presence of fish; *Daphnia* can recognize the presence of the fish chemically and can show adaptive plasticity responses; these were thus also quantified, resulting in reaction norms for 14 traits of 36 clones). Out of 14 traits studied, 13 traits showed significant evolutionary changes, and these changes were in line with expectations under the assumption of adaptive evolution. For example, in the case of diel vertical migration measured as phototactic

behaviour, adaptive evolution would involve a change towards becoming more negatively phototactic, as residing deeper in the water column is a well-documented and effective defence against predation by visually hunting fish. This is also what we observed: during the first transition from no-fish to high-fish, we observed a significant evolution of phenotypic plasticity. More specifically, in the high-fish predation subpopulation most clones show a strong and significant phenotypic plasticity towards more negative phototaxis (i.e. safer behaviour) in the presence of fish smell, whereas most clones in the pre-fish subpopulation did not show such a phenotypic plasticity response. This thus reflects adaptive evolution through genetic changes in phenotypic plasticity. In the second transition, from high to reduced fish predation, we did not see an evolution of phenotypic plasticity, but observed a genetic shift in mean trait value to less negatively phototactic behaviour. So, the clones remain responsive to fish but in general behave less safe - in line with expectations that there was still fish present in the habitat, but in less high abundances than before. Similar adaptive changes were observed for size at maturity, with animals genetically becoming smaller in the high fish subpopulation and partly reversed that response to become somewhat larger again in the reduced-fish subpopulation. There is no space in the current manuscript to go in depth into these trait changes, but key is that we observed adaptive evolution in multiple traits and that these trait changes were (partly) reversible. The allele frequency changes that we see at the level of the genome are very much in line with these trait changes. We here observed reversals in 99% (1753 SNPs) of SNPs that show significant changes in allele frequency in both the pre-fish to high-fish and the high-fish to reduced-fish transitions, representing a total of 1771 SNPs. This percentage of overall reversal is more than expected by chance based on a permutation test (p -value $< 1e-4$) (here we followed the approach described in "Section S6 of Buffalo & Coop (2020) - Estimating the genome-wide contribution of selection to temporal allele frequency change, PNAS 117 (34) 20672-20680)". Another line of evidence on the adaptive nature of the observed changes in allele frequencies is provided by the fact that all three significantly overrepresented pathways are in the pathway class of "Environmental Information Processing; Signaling molecules and interaction" and were validated by the spatial analysis.

So, we indeed did these common garden experiments, but they were published prior to our genomic analysis. Our genomic analyses were actually inspired by the striking results we obtained concerning the extensive adaptive evolution in our common garden study focusing on traits. Given these earlier data on adaptive evolution and given that our full-genome sequencing analysis captures all genomic variation and demonstrates, similar to the changes in trait values, extensive and repeatable changes through time, we feel it is safe to link the genomic responses to the adaptive trait evolution. We can, however, not carry out a GWAS analysis, because our common garden experiment on 14 different traits involved a total of 36 clones, and this number does not allow a statistically powerful GWAS analysis.

Inspired by this comment, we did implement some changes and added the explanation as supplementary text in the revised version of the manuscript to make the link to the previous common garden study on trait evolution more clear.

Minor comments:

line 46-47: selective sweeps are actually the reason we can often find the loci targeted by selection

A29. The reviewer is correct in stating that selective sweeps are responsible for the "footprint" of natural selection, pointing to a locus or loci within a defined interval of the genome as its target. However, apart from the advantage of selective sweeps at detecting a local region that is influenced by selection, they also conceal the identity of the actual target. Our modified paragraph is improved by presenting this dichotomy more clearly.

line 47-48: I am not sure why a failure to predict if populations evolve in reverse or adapt to other things is the fault of the sweep

A30. This statement's original intention was to hint on the potential for observing loci within swept blocks within our temporal study that follow separate evolutionary trajectories if populations evolve in reverse. To improve overall clarity of the introduction, this statement is now removed from the paragraph.

line 81: given N_e and the number of generations that has passed you should be able to predict the expected F_{st} due to drift alone

*A31. Based on Bayesian simulations, we found negligible effect of drift (Fig. R5). The drift component $F_s = F_{st} - F_t$ was identified using the TAFT software as described in "Testing temporal changes in allele frequencies: a simulation approach", *Genet. Res., Camb.* (2010), 92, pp. 309–320." by Edson Sandoval-Castellanos. Overall, for all SNPs and outlier SNPs in both transitions (pre-fish to high fish and high-fish to reduced fish), a maximum F_{st} value of 0.025 can be attributed to drift only. These values are comparable to mean F_{st} values for all loci in pre-fish to high-fish (mean F_{st} =0.027) and high-fish to reduced-fish transition (mean F_{st} =0.010). Furthermore, linear relationship between F_{st} and F_t components for all loci and outlier loci further strengthens our claim that there is little or no detectable contribution of genetic drift in structuring the genome-wide polymorphisms. The results are based on 10,000 simulations using TAFT software.*

Fig R5. A) Drift component i.e. ($F_{st}-F_t$) for all genotyped SNPs in OHZ temporal population for pre-fish vs high-fish comparison. B) Relationship between F_{st} and F_t for all loci in pre-fish vs high-fish comparison. C) Drift component i.e. ($F_{st}-F_t$) for 1109 SNPs belonging to the divergent SNPs that showed significant changes in allele frequencies in the pre-fish to high-fish transition and a significant reversal during the high-fish to reduced-fish transition, excluding hitchhiked loci in the OHZ population for the pre-fish vs high-fish comparison. D) Relationship between F_{st} and F_t for 1109 SNPs belonging to the divergent SNPs that showed significant changes in allele frequencies in the pre-fish to high-fish transition and a significant reversal during the high-fish to reduced-fish transition excluding hitchhiked loci in the OHZ population for pre-fish vs high-fish comparison. E) Drift component i.e. ($F_{st}-F_t$) for all genotyped SNPs in OHZ temporal population for high-fish vs reduced-fish comparison. F) Relationship between F_{st} and F_t for all loci in high-fish vs reduced-fish comparison. G) Drift component i.e. ($F_{st}-F_t$) for 1109 SNPs belonging to the divergent SNPs that showed significant changes in allele frequencies in the pre-fish to high-fish transition and a

significant reversal during the high-fish to reduced-fish transition excluding hitchhiked loci in the OHZ population for high-fish vs reduced-fish comparison. H) Relationship between F_{st} and F_t for 1109 SNPs belonging to the divergent SNPs that showed significant changes in allele frequencies in the pre-fish to high-fish transition and a significant reversal during the high-fish to reduced-fish transition excluding hitchhiked loci in the OHZ population for high-fish vs reduced-fish comparison.

line 84: I am not sure nearly 10% of the genome is a "small fraction"

A32. See answer A7 to reviewer 1

line 91-98: could you perform permutation tests to statistically test whether the number of reversals is significant? (see Section S6 of Buffalo & Coop 2020 bioRxiv 798595)

A33. We estimated the null distribution by randomly permuting the transition phase labels, (i.e., pre-fish, high-fish, and reduced-fish) and alleles at the locus-level (724321 SNPs) and recalculating the number of reversals. We found that the observed number of reversals (1753) is significantly larger than the mean of 10000 runs of permutations (994), with p -value $< 1e-4$. (here we followed the approach described in described in "Section S6 of Buffalo & Coop (2020) - Estimating the genome-wide contribution of selection to temporal allele frequency change, PNAS 117 (34) 20672-20680). We implemented the results on L129-L131.

lines 105 and 108: I find it clearer to differentiate sweeps (increase in frequency due to positive selection) from hitchhiking (increase in frequency due to linkage with a sweeping allele)

A34. We now updated the text for more clarity "However, because of genetic hitchhiking, not all genes with polymorphisms that show reversals are predicted to be the direct targets of natural selection (Selective sweep: increase in frequency of due to positive selection; genetic hitchhiking: increase in frequency due to linkage with a sweeping allele)"

line 118: aren't the islands of divergence in the second transition measured relative to the high fish population, and therefore only 10 years has elapsed (rather than 16)?

A35. The reviewer is right. The overlap reflects a 16-year period perspective, but the smaller islands in the second transition reflect a period of 10 years. We changed this in the text.

line 179: a rarefaction curve within a population would be interesting too, as it would say whether the variation is already there locally, without the need for migration. I guess the low F_{st} suggests all the variation is indeed already in each population

A36. We actually did the rarefaction curve also within population OHZ; see line L772 in Figure4A. The pattern is indeed very similar, with most of the variation already present in the pre-fish population.

Below we do show the results of a rarefaction analysis for the OHZ population when using immigrants from each of the spatial populations separately (seeded by regional genotypes). It is clear that all populations would be good sources for the genetic variation of population OHZ. Population OHN, which is a neighboring pond, results in the steepest accumulation of genetic variation (Fig. R6).

Fig. R6 The proportion of polymorphic positively selected SNPs in the OHZ population that were also found in the spatial dataset (N = 1003; i.e. 90.4 % of the loci from Fig. 4A) that is polymorphic in a simulated founding propagule as a function of an increasing number of individual genotypes in the founding module from each of the 12 sampled spatial populations (i.e six populations from ponds without fish (NF) and six populations from ponds with fish (F))

line 199: again confused about SNP filtering -- you would include SNPs that fixed?

A37. The fixed differences were not removed during the filtering process. Please see Fig R3 and answer A24, indicating that there were no fixed differences in each pairwise population comparison.

line 280: so you only include SNPs in genes? this seems like a major understated decision as the paper discusses the genome, the percent of the genome under selection, etc

A38. The data on the whole genome was used. For clarity in the current version of the manuscript we edited the sentence to "such that each polymorphic SNP contained within genic regions could be unambiguously assigned to only one gene locus, thereby removing uncertainties attributed to sequence reads mapping to paralogs and to overlapping genes coding on alternative strands of DNA." This filtering step did not affect the SNPs that were falling outside genic regions.

line 313-314: unclear -- these SNPs changed allele frequency in both transitions?

A39. Yes, for clarity we changed the sentence "1771 SNPs showed significant allele frequency changes in both transitions, most of them also showing a significant reversal in the second transition." See lines L439-440

line 336: Gaussian $\log_{10}(F_{st})$ seems like an odd choice given F_{st} can be zero (what did you do then?) and is maximized at 1 (while the Gaussian goes to infinity). I see Soria-Carrasco et al (cited) use a logit, which does not have these problems. Why the change?

A40. We choose $\log_{10}(F_{st}+1)$ to accommodate the negative values generated by Weir and Cockerham 1984 F_{st} .

We use the script in this repository: <https://github.com/marqueda/HMM-detection-of-genomic-islands>, which was described in the following publications:

Hofer T, M Foll & L Excoffier (2012) Evolutionary forces shaping genomic islands of population differentiation in humans. BMC Genomics 13:107.

Marques DA, K Lucek, MP Haesler, AF Feller, JI Meier, CE Wagner, L Excoffier & O Seehausen (2016b) Genomic landscape of early ecological speciation initiated by selection on nuptial color. Molecular Ecology 26: 7-24.

Sorio-Carrasco V, Z Gompert, AA Comeault, TE Farkas, TL Parchman, JS Johnston, CA Burkle, JL Feder, J Bast, T Schwander, SP Egan, BJ Crespi & P Nosil (2014) Stick insect genomes reveal natural selection's role in parallel speciation. Science 344: 738-742.

line 336-337: how can you initialize like this before you know which SNPs are in which level?

A41. The Gaussian distributions of state 1, 2, and 3 are initialized using the mean and standard deviation (sd) of the F_{st} values smaller than the 0.5 quantile, between the 0.5 and 0.99 quantiles, and larger than the 0.99 quantile, respectively. The Hidden Markov model is randomly initialised. These parameters are then optimised by using 1,000 runs of the Baum-Welch algorithm to maximise the model likelihood. After the HMM model is trained, the state of each SNP is reconstructed using the Viterbi algorithm.

Data vs. Emission & Transition Probabilities

Fig. R7. The distribution of $\log_{10}(F_{ST}+1)$ and three gaussian distributions defining low (state 1; blue line), moderate (state 2; black line) and high differentiation (state 3; red line) category SNPs. These three Gaussian distributions of the states were initialized based on mean and standard deviation (sd) of $\log_{10}(F_{ST}+1)$ values smaller than the 0.5 quantile (state 1), between the 0.5 and 0.99 quantiles (state 2), and larger than the 0.99 quantile (state 3), respectively. The same gaussian model was used in both temporal transitions.

line 338: was the same Gaussian model (ie the mean and variance of $\log_{10}(F_{ST})$ within each level) used for both temporal transitions? If not, is it fair to compare levels of differentiation across the transitions?

A42. Yes, it is the same gaussian model. Please have a look at the above figure (Fig. R7) which was generated from the $\log_{10}(F_{ST}+1)$ values from both transition phases.

line 340: "estimated parameters" has some grammar issues, maybe "we estimated the parameters"

A43. We implemented this suggestion

line 369: in -> as a

A44. Implemented

line 377: genetic -> adaptive genetic (otherwise theta tells you the answer)

A45. Implemented

line 380: say where 2204 comes from

A46. In the previous version of the manuscript 2204 SNPs belonging to the divergent SNPs that showed significant changes in allele frequencies in the pre-fish to high-fish transition and a significant reversal during the high-fish to reduced-fish transition excluding hitchhiked loci in the OHZ population. For clarity, we changed it in current version to “1109 SNPs belonging to the divergent SNPs that showed significant changes in allele frequencies in the pre-fish to high-fish transition and also a significant reversal during the high-fish to reduced-fish transition excluding hitchhiked loci in the OHZ population”

line 394: missing a ")"

A47. Implemented

line 483: underwent a rapid -> underwent rapid

A48. Implemented

fig 2: why do some alleles start below 0.05 (despite filtering)? Do I misunderstand the filtering or is this a result of the binomial model of frequency you use?

A49. The filtering of allele frequency was performed using the 0.05 cut-off, but based on all sub-populations pooled. Hence, allele frequencies can be lower than 0.05 in specific subpopulations.

Supplementary material:

line 66: "pair" -> "triplet"

A50. Implemented

line 74: how can you see these three categories in the plot? It seems like only "high" is coloured? (In general it is not clear why the moderate class exists.)

A51. The reviewer is right. It was an error in the figure legend which has now been corrected with the following statement “Hidden Markov Model SNP categories representing genomic regions of high genetic differentiation”. We were interested in only identifying regions of high genetic differentiation using HMM, the other regions of low divergence might represent other unknown mechanisms and require further investigation.

REVIEWER COMMENTS

Reviewer #1 (Remarks to the Author):

The authors have integrated all of my comments. I have no further comments on the manuscript. I really like the revised discussion.

Reviewer #2 (Remarks to the Author):

The authors use whole genome resequencing of *Daphnia magna* populations over time (with cores) and space (across ponds) to explore how genetic variation is structured and responds to selection (fish predation). In their focal population, where they have sequences before, during, and after high fish predation, they find that ~5% of SNPs change frequency more than expected by drift alone, and ~5% of these show a significant reversal in the direction of change once predation is relaxed. Using a Hidden Markov Model, they show that 80% of these reversal SNPs are in genomic islands of high genetic differentiation (between timepoints) and that there are likely up to 342 genes under selection by fish predation (interestingly, 92% of these genes are in islands of high differentiation between populations with and without fish predation). Nearly all the genetic variation observed is contained in only 12 individuals before exposure to fish, is retained in the 12 individuals after exposure to fish, and can be built up with only ~5 individuals from surrounding populations. All this leads the authors to conclude that there is rampant standing genetic variation in this system which allows rapid, reversible evolution.

I thank the authors for their revisions. Based on their response I now understand better how SNPs were filtered (below 5% after pooling subpopulations, line 407-408), which remedies my concern about new mutations. I am also happy to see a much clearer explanation of how they determined which SNPs significantly changed frequency, which now incorporates drift (N_e), and so my concerns there are also relieved. And I am happy to see the perturbation analysis used to determine if the number of reversals is statistically significant. My main comment about their response is that I think the number of genes in the overlapping high-differentiation regions (~1 per region, as given in the response) should also be given in the main text, as that helps me believe that most of the putatively selected genes are not just hitchhikers.

My remaining comments are:

1) There is some confusion/inconsistency around the words "population" and "subpopulation". At first it appears that different times within a pond will be considered subpopulations and that different ponds will be considered populations. This would be OK if consistent (although I would rather 'temporal subpopulation' be used at every instance, rather than just 'subpopulation', as I tend to think of a single population or subpopulation existing through time), but becomes inconsistent throughout.

2) I would like more caution to be added to the language around adaptive SNPs and genes. For example

line 161 says "allows us to identify", line 174 says "genes that were direct targets of selection (ie., excluding hitchhiked genes)", line 206 says "genomic islands under directional selection", line 215 says "polymorphisms under selection", line 226 says "genes under directional selection ... (ie., excluding hitchhiked genes)", line 237 says "selected genomic regions", On a related note, line 251 says "putative[ly] adaptive SNPs", but I'd argue you can only call one SNP per region putatively adaptive as there is no way to tell which is the driver and which are the hitchhikers. Perhaps "potentially" would be a better word?

3) I think more explanation needs to be added to the finding that evolution appears to be reversible, because genetic variation was not eroded, in this instance. I don't think this, by itself, is all that interesting. As the authors say (line 215-216), none of the potentially selected SNPs reached fixation during this brief episode of selection. So the explanation for why genetic variation was not eroded could be as simple as "selection acted for too short a time to cause fixation". Obviously a more interesting alternative is that the traits under selection are highly polygenic, with weak selection acting on any one SNP, so that the genetic architecture is what makes evolution reversible. So then we are in some sweet spot where there are enough loci under selection that none fix, but there are few enough loci under selection that we can detect selection's effect on their allele frequencies?

4) The authors mention a generation time of 1 year, but it would help me (who knows little about *Daphnia*) to know more about how often sexual reproduction and recombination happen in this system, as well as how often asexual reproduction occurs. Does the 1 year just refer to sexual reproduction, with many asexual generations within a year? This changes our expectations for the amount of drift we expect from one time sample to the next (so should be involved in the Waples test), and would help the reader put things into context.

5) I realize the other reviewer asked for more talk about conservation in the Discussion, but it is unclear to me what this study, with N_e greater than a million for a single pond (not to mention the census size) and no demographic data, has to do with evolutionary rescue (line 290). It seems simpler and more reasonable to just stick to talking about rapid adaptation and the maintenance of diversity. Obviously populations with huge sizes are more likely to persist, so this doesn't strike me as a particularly exciting path to go down in a short discussion. Another topic might be to talk about the ability to colonize new habitats, perhaps during range shifts caused by climate change?

6) I think the result that 92% of the genes in putatively adaptive islands across time are also in islands across space is very interesting and could use more emphasis and explanation. This changes my interpretation of the temporal results from one where all the islands of differentiation are created by sweeping beneficial alleles on many backgrounds to one where one or a few fish-adapted clones increase in frequency (and potentially have only a locus or two that aids in protection from fish, as all islands are in LD with each other on that background). But if the former explanation is correct, then this observation strengthens the findings as it suggests evolution is repeatable, and hence probably adaptive.

line 143-144: This addition seems a bit awkward to me, e.g., a sweep is defined before it is mentioned. Why not just define hitchhiking after it's used?

line 146: There still seems to be some confusion around the metaphor. My understanding is that a beneficial allele sweeps to high frequency, and the nearby ancestral variation hitchhikes with it. So I don't think "sweeps ancestral variation" is in keeping with tradition.

line 149-158: I'm not asking the authors to do this, but it struck me while reading this time that it would be neat to see the islands of high differentiation compared with classic measures of selection using contemporary samples alone (e.g., π and Tajima's D). I wonder if the two methods would line up?

line 157: The small island sizes during reduced fish predation is interesting. The authors claim this is because there has been more time for recombination to occur, but I'm not convinced this works out because there has also been more time for selection to act (creating more linkage disequilibrium). Alternative hypotheses include weaker selection (because of a smaller change in fish density). Whatever tack the authors take (convincing the reader their argument makes sense, ie is consistent with theory, or listing alternative hypotheses), I think the language should be softened and the cause is unknown.

line 266: The authors seem to be implying that documenting evolution to an in situ environmental change is unique. But there are lots of studies that do that (e.g., Reid et al 2016 Science, Campbell-Staton et al 2017 Science, etc). I think this statement should be clarified so as not to claim more novelty than appropriate (I imagine adding something about temporal samples of whole genomes would help).

fig s2: Needs letters on the panels

line 135 of SOM: It should be specified that these are mean trait and genotypic values (there is a random environmental effect on every individual trait, but you can hope these cancel out when you take the mean so that you can measure the genotypic mean).

REVIEWER COMMENTS

Reviewer #1 (Remarks to the Author):

The authors have integrated all of my comments. I have no further comments on the manuscript. I really like the revised discussion.

Reviewer #2 (Remarks to the Author):

The authors use whole genome resequencing of *Daphnia magna* populations over time (with cores) and space (across ponds) to explore how genetic variation is structured and responds to selection (fish predation). In their focal population, where they have sequences before, during, and after high fish predation, they find that ~5% of SNPs change frequency more than expected by drift alone, and ~5% of these show a significant reversal in the direction of change once predation is relaxed. Using a Hidden Markov Model, they show that 80% of these reversal SNPs are in genomic islands of high genetic differentiation (between timepoints) and that there are likely up to 342 genes under selection by fish predation (interestingly, 92% of these genes are in islands of high differentiation between populations with and without fish predation). Nearly all the genetic variation observed is contained in only 12 individuals before exposure to fish, is retained in the 12 individuals after exposure to fish, and can be built up with only ~5 individuals from surrounding populations. All this leads the authors to conclude that there is rampant standing genetic variation in this system which allows rapid, reversible evolution.

I thank the authors for their revisions. Based on their response I now understand better how SNPs were filtered (below 5% after pooling subpopulations, line 407-408), which remedies my concern about new mutations. I am also happy to see a much clearer explanation of how they determined which SNPs significantly changed frequency, which now incorporates drift (N_e), and so my concerns there are also relieved. And I am happy to see the perturbation analysis used to determine if the number of reversals is statistically significant. My main comment about their response is that I think the number of genes in the overlapping high-differentiation regions (~1 per region, as given in the response) should also be given in the main text, as that helps me believe that most of the putatively selected genes are not just hitchhikers.

A1. We thank the reviewer for her/his positive assessment of our manuscript and revision. Following the suggestion of the reviewer, we included the number of genes in the overlapping high-differentiation regions in the main text of the manuscript (L156-L160).

My remaining comments are:

1) There is some confusion/inconsistency around the words "population" and "subpopulation". At first it appears that different times within a pond will be considered subpopulations and that different ponds will be considered populations. This would be OK if consistent (although I would rather 'temporal subpopulation' be used at every instance, rather than just 'subpopulation', as I tend to think of a single population or subpopulation existing through time), but becomes inconsistent throughout.

A2. We thank the reviewer for pointing out this inconsistency. We changed “subpopulation” to “temporal subpopulation” throughout the main text and supplementary text. We also verified the consistency of the use of “population” for the spatial populations throughout the text.

2) I would like more caution to be added to the language around adaptive SNPs and genes. For example line 161 says "allows us to identify", line 174 says "genes that were direct targets of selection (ie., excluding hitchhiked genes)", line 206 says "genomic islands under directional selection", line 215 says "polymorphisms under selection", line 226 says "genes under directional selection ... (ie., excluding hitchhiked genes)", line 237 says "selected genomic regions", On a related note, line 251 says "putative[ly] adaptive SNPs", but I'd argue you can only call one SNP per region putatively adaptive as there is no way to tell which is the driver and which are the hitchhikers. Perhaps "potentially" would be a better word?

A3. We agree with the reviewer, we added “potential(ly)” at several places in main text including L149, L155, L163, L166, L183, L192, L201, L212, L224, L249, L478, L486, L710, L716, and lines 96, 106, and 108 of SOM.

3) I think more explanation needs to be added to the finding that evolution appears to be reversible, because genetic variation was not eroded, in this instance. I don't think this, by itself, is all that interesting. As the authors say (line 215-216), none of the potentially selected SNPs reached fixation during this brief episode of selection. So the explanation for why genetic variation was not eroded could be as simple as "selection acted for too short a time to cause fixation". Obviously a more interesting alternative is that the traits under selection are highly polygenic, with weak selection acting on any one SNP, so that the genetic architecture is what makes evolution reversible. So then we are in some sweet spot where there are enough loci under selection that none fix, but there are few enough loci under selection that we can detect selection's effect on their allele frequencies?

A4. In principle both explanations would indeed be possible. The former one, however, is perhaps somewhat less likely, given that we actually observed a similar level of polymorphism in all spatial populations too. So, if the explanation would be that selection acted for a too short time to cause fixation, we should conclude that this would be rather general across populations and settings. We therefore also think that the alternative explanation is more likely – the fact that there is polygenic inheritance, with rather weak selection on each SNP – to explain the lack of genetic erosion in our data. Following the suggestion of the reviewer, we highlight both possibilities from L247-L254 (selection acting too short to cause fixation vs. polygenic nature of the traits with an intermediate number of loci of moderate effect).

4) The authors mention a generation time of 1 year, but it would help me (who knows little about Daphnia) to know more about how often sexual reproduction and recombination happen in this system, as well as how often asexual reproduction occurs. Does the 1 year just refer to sexual reproduction, with many asexual generations within a year? This changes our expectations for the amount of drift we expect from one time sample to the next (so should be involved in the Waples test), and would help the reader put things into context.

A5. From year to year, the animals hatch at the start of the growing season from the dormant eggs resulting from sexual reproduction, and at the end of the growing season there is one round of sexual reproduction leading to the production of dormant stages that overwinter. We used a generation

time of one year because the cyclical parthenogenetic reproduction mode of Daphnia involves in most cases one generation of sexual reproduction per year (in the study region: mostly in fall to go dormant during the winter). Daphnia during the growing season has multiple parthenogenetic generations – with approximately two parthenogenetic generations per month at summer temperatures. So, in total one might estimate 10-15 parthenogenetic generations per year and one sexual generation.

There are good reasons to use the sexual cycle as the relevant generation time. First, it is during sexual reproduction that recombination occurs. Secondly, while several generations of parthenogenetic reproduction during the growing season might lead to strong clonal selection, in practice we see very low levels of year-to-year genetic differentiation for genome-wide markers when working with dormant eggs collected from sediment cores (Orsini et al 2016, Molecular Ecology; and this study). This low level of genetic drift is due to a combination of very large population sizes and the fact that while competition among clones is strong thus leading to clonal selection. The populations start with a high number of clones (in the millions) and at the end of the growing season there might still be 10,000 clones left, each represented by on average, for instance, 1000 individuals. It is easy to imagine that with the high level of genetic variation that we observed sequencing 12 clones in a given temporal subpopulation in the present study, that random mating of 10,000 clones each represented by many individuals will, for neutral markers, yield very stable allele frequencies through time.

In response to the reviewer's comment, we performed the Waples test both assuming one or ten generations per year, and the results were nearly identical. More specifically on average across all 1,753 SNPs that show significant changes in allele frequency in the pre-fish to high-fish transition and also show a significant (Waples test, P-values < 0.01) partial reversal of this change during the high-fish to reduced-fish transition, the influence of increasing the number of generations times ten on the p-values is less than then 0.06% for pre-fish to high-fish and 0.13% for high-fish to reduced-fish transition. This implies that the number of SNPs that would change status from outlier to non-outlier or vice versa is negligible.

5) I realize the other reviewer asked for more talk about conservation in the Discussion, but it is unclear to me what this study, with N_e greater than a million for a single pond (not to mention the census size) and no demographic data, has to do with evolutionary rescue (line 290). It seems simpler and more reasonable to just stick to talking about rapid adaptation and the maintenance of diversity. Obviously populations with huge sizes are more likely to persist, so this doesn't strike me as a particularly exciting path to go down in a short discussion. Another topic might be to talk about the ability to colonize new habitats, perhaps during range shifts caused by climate change?

A6. We partly disagree with the reviewer. First, even organisms with huge census and effective population sizes can go extinct (e.g the passenger pigeon; <https://science.sciencemag.org/content/358/6365/951>). Also, Daphnia populations can rapidly go extinct. We don't have detailed demographic data, but in Figure 1 we present resting egg density data which show a decline and a recovery phase, as expected under evolutionary rescue (while not being a proof of evolutionary rescue, as recovery can also in part be related to a relaxation of the selection pressure). Second, our study is in our opinion relevant to conservation genetics because of the link between local and regional diversity. We observe high standing genetic diversity, the maintenance of which is perhaps indeed not surprising giving the large effective population sizes. However, this standing genetic variation needs first to build up, and this is through immigration from regional populations. The striking fact that only a few immigrants are needed to establish this high

level of standing genetic variation is strongly dependent on the fact that all regional populations are harbouring such a high level of standing genetic variation. This led us to the statement that to maintain genetic variation one should not only ensure that local populations have a sufficiently large effective population size, but also ensure that this is the case across a large portion of the populations regionally. Because of this relevance, we did not delete this short discussion from the main text.

6) I think the result that 92% of the genes in putatively adaptive islands across time are also in islands across space is very interesting and could use more emphasis and explanation. This changes my interpretation of the temporal results from one where all the islands of differentiation are created by sweeping beneficial alleles on many backgrounds to one where one or a few fish-adapted clones increase in frequency (and potentially have only a locus or two that aids in protection from fish, as all islands are in LD with each other on that background). But if the former explanation is correct, then this observation strengthens the findings as it suggests evolution is repeatable, and hence probably adaptive.

A7. We agree that the link between temporal and spatial variation is important and adds confidence in our results. We do not agree with the suggestion of a few fish-adapted clones that increase in frequency, as then we should have seen more year-to-year variation whereas the observed F_{st} values are very low. Our interpretation rather is the following: While there is strong selection among clones during the growing season, this selection does not result in just a few clones surviving at the end of the growing season but rather a few thousands of clones that have a favourable combination of traits (in brief, we expect a reduction in number of clones from say 100 million to a few thousand). We do agree that our results point towards repeatable (at least partially) and adaptive evolution, while not leading to strong year-to-year variation and strong genetic erosion. To put it in other words, key to the patterns that we observe is the very high clonal diversity which represents an equally large number of combinations of the high standing genetic variation at the level of loci.

line 143-144: This addition seems a bit awkward to me, e.g., a sweep is defined before it is mentioned. Why not just define hitchhiking after it's used?

A8. We defined hitchhiking just after it is used (L128).

line 146: There still seems to be some confusion around the metaphor. My understanding is that a beneficial allele sweeps to high frequency, and the nearby ancestral variation hitchhikes with it. So I don't think "sweeps ancestral variation" is in keeping with tradition.

A9. We changed the wording to resolve the confusion.

line 149-158: I'm not asking the authors to do this, but it struck me while reading this time that it would be neat to see the islands of high differentiation compared with classic measures of selection using contemporary samples alone (e.g., π and Tajima's D). I wonder if the two methods would line up?

A10. We thank the reviewer for the suggestion. Our data would indeed offer some opportunities to verify population genetic theory by comparing patterns of selection over time with summary statistics in the contemporary samples. We considered this, however, it is outside of the scope of the present study.

line 157: The small island sizes during reduced fish predation is interesting. The authors claim this is because there has been more time for recombination to occur, but I'm not convinced this works out because there has also been more time for selection to act (creating more linkage disequilibrium). Alternative hypotheses include weaker selection (because of a smaller change in fish density). Whatever tack the authors take (convincing the reader their argument makes sense, ie is consistent with theory, or listing alternative hypotheses), I think the language should be softened and the cause is unknown.

A11. We indeed agree there could be alternative hypotheses. Therefore, we softened the language by using "likely".

line 266: The authors seem to be implying that documenting evolution to an in situ environmental change is unique. But there are lots of studies that do that (e.g., Reid et al 2016 Science, Campbell-Staton et al 2017 Science, etc). I think this statement should be clarified so as not to claim more novelty than appropriate (I imagine adding something about temporal samples of whole genomes would help).

A12. We slightly edited this sentence, also emphasizing that we use whole genome data in a temporal setting (L238).

fig s2: Needs letters on the panels

A13. We thank the reviewer for the pointer. We added the panels and updated the Fig. S2.

line 135 of SOM: It should be specified that these are mean trait and genotypic values (there is a random environmental effect on every individual trait, but you can hope these cancel out when you take the mean so that you can measure the genotypic mean).

*A14. We now specify this information. We do note that the data in these figures are based on three to five replicated observations/clone, and 12 clones/population. The variation among genotypes is actually high within temporal subpopulations, hence the heritability of these traits is also high (Stoks, R., Govaert, L., Pauwels, K., Jansen, B. & De Meester, L. Resurrecting complexity: the interplay of plasticity and rapid evolution in the multiple trait response to strong changes in predation pressure in the water flea *Daphnia magna*. *Ecol Lett* 19, 180-190, (2016).)*

REVIEWERS' COMMENTS

Reviewer #2 (Remarks to the Author):

Thanks to the authors for dealing with another round of questioning. I especially appreciate the added discussion of short term selection vs polygenicity in making evolution reversible, which I hope will benefit readers. I think the responses regarding generation times and evolutionary rescue make sense and that it might help the reader if a brief summary of that info was also presented in the paper.

Reviewer #2 (Remarks to the Author):

Thanks to the authors for dealing with another round of questioning. I especially appreciate the added discussion of short term selection vs polygenicity in making evolution reversible, which I hope will benefit readers. I think the responses regarding generation times and evolutionary rescue make sense and that it might help the reader if a brief summary of that info was also presented in the paper.

We thank the reviewer for her/his constructive comments and suggestions during the whole process. With respect to the reviewer's final recommendation that a brief summary of our responses regarding generation times and evolutionary rescue be added to the discussion, we debated on how best to adapt the discussion, and decided to modify the final sentences of the paragraph on L252-L261, beginning with "Furthermore, our data suggest that...". In this paragraph, we list the key features leading to the maintenance of standing genetic variation. In addition to this paragraph also the final paragraph of the main manuscript (L292-298) illustrates the relevance of our paper to conservation genetics, as it is clear from our data that local evolutionary potential may often require conservation at the landscape level, ensuring sufficient standing genetic variation at multiple sites that can represent sources for colonists and immigrants.